# Integrating influenza antigenic dynamics with molecular evolution

Trevor Bedford[1]*[†], Marc A Suchard[2,3,4], Philippe Lemey[5], Gytis Dudas[1], Victoria Gregory[6], Alan J Hay[6], John W McCauley[6], Colin A Russell[7,8], Derek J Smith[7,9,10], Andrew Rambaut[1,11,12]

[1]Institute of Evolutionary Biology, University of Edinburgh, Edinburgh, United Kingdom; [2]Department of Biomathematics, David Geffen School of Medicine, University of California, Los Angeles, Los Angeles, United States; [3]Department of Human Genetics, David Geffen School of Medicine, University of California, Los Angeles, Los Angeles, United States; [4]Department of Biostatistics, UCLA Fielding School of Public Health, University of California, Los Angeles, Los Angeles, United States; [5]Department of Microbiology and Immunology, Rega Institute, Katholieke Universiteit Leuven, Leuven, Belgium; [6]Division of Virology, MRC National Institute for Medical Research, London, United Kingdom; [7]WHO Collaborating Centre for Modeling, Evolution and Control of Emerging Infectious Diseases, University of Cambridge, Cambridge, United Kingdom; [8]Department of Veterinary Medicine, University of Cambridge, Cambridge, United Kingdom; [9]Centre for Pathogen Evolution, Department of Zoology, University of Cambridge, Cambridge, United Kingdom; [10]Department of Virology, Erasmus Medical Centre, Rotterdam, Netherlands; [11]Fogarty International Center, National Institutes of Health, Bethesda, United States; [12]Centre for Immunity, Infection and Evolution, University of Edinburgh, Edinburgh, United Kingdom

*For correspondence: tbedford@fhcrc.org

[†]Present address: Vaccine and Infectious Disease Division, Fred Hutchinson Cancer Research Center, Seattle, United States

Competing interests: The authors declare that no competing interests exist.

**Abstract** Influenza viruses undergo continual antigenic evolution allowing mutant viruses to evade host immunity acquired to previous virus strains. Antigenic phenotype is often assessed through pairwise measurement of cross-reactivity between influenza strains using the hemagglutination inhibition (HI) assay. Here, we extend previous approaches to antigenic cartography, and simultaneously characterize antigenic and genetic evolution by modeling the diffusion of antigenic phenotype over a shared virus phylogeny. Using HI data from influenza lineages A/H3N2, A/H1N1, B/Victoria and B/Yamagata, we determine patterns of antigenic drift across viral lineages, showing that A/H3N2 evolves faster and in a more punctuated fashion than other influenza lineages. We also show that year-to-year antigenic drift appears to drive incidence patterns within each influenza lineage. This work makes possible substantial future advances in investigating the dynamics of influenza and other antigenically-variable pathogens by providing a model that intimately combines molecular and antigenic evolution.

## Introduction

Seasonal influenza infects between 10% and 20% of the human population every year, causing an estimated 250,000 to 500,000 deaths annually (*Influenza Fact sheet, 2009*). Although individuals develop long-lasting immunity to particular influenza strains after infection, antigenic mutations to the influenza virus genome result in proteins that are recognized to a lesser degree by the human immune system, leaving individuals susceptible to future infection. The influenza virus population continually

**eLife digest** Every year, seasonal influenza, commonly called flu, infects up to one in five people around the world, and causes up to half a million deaths. Even though the human immune system can detect and destroy the virus that causes influenza, people can catch flu many times throughout their lifetimes because the virus keeps evolving in an effort to avoid the immune system. This antigenic drift—so-called because the antigens displayed by the virus keep changing—also explains why influenza vaccines become less effective over time and need to be reformulated every year.

It is possible to determine which antigens are displayed by a new strain of the virus by observing how blood samples that respond to known strains respond to the new strain. This information about the "antigenic phenotype" of the virus can be plotted on an antigenic map in which strains with similar antigens cluster together. Gene sequencing has shown that there are four subtypes of the flu virus that commonly infect people; but the relationship between changes in antigenic phenotype and changes in gene sequences of the influenza virus is poorly understood.

Bedford et al. have now developed an approach to combine antigenic maps with genetic information about the four subtypes of the human flu virus. This revealed that the antigenic phenotype of H3N2—a subtype that is becoming increasingly common—evolved faster than the other three subtypes. Further, a correlation was observed between antigenic drift and the number of new influenza cases per year for each flu strain. This suggests that knowing which antigenic phenotypes are present at the start of flu season could help predict which strains of the virus will predominate later on.

The work of Bedford et al. provides a useful framework to study influenza, and could help to pinpoint which changes in viral genes cause the changes in antigens. This information could potentially speed up the development of new flu vaccines for each flu season.

evolves in antigenic phenotype in a process known as antigenic drift. A large proportion of the disease burden of influenza stems from antigenic drift, which allows individuals to be infected multiple times throughout their lives. Although influenza vaccines may lack efficacy for a variety of reasons (*Osterholm et al., 2012*), antigenic drift causes efficacy of a fixed vaccine formulation against circulating viruses to decline over time. A thorough understanding of the process of antigenic drift is essential to public health efforts to control mortality and morbidity through the use of a seasonal influenza vaccine.

Before 2009, there were four major lineages of influenza circulating within the human population: the H3N2 and H1N1 subtypes of influenza A, and the Victoria and Yamagata lineages of influenza B. In the case of influenza A, subtypes A/H3N2 and A/H1N1 refer to the genes, hemagglutinin (H or HA) and neuraminidase (N or NA), that are primarily responsible for the antigenic character of a strain. In the case of influenza B, Victoria (B/Vic) and Yamagata (B/Yam) refer to antigenically distinct lineages which diverged from a single lineage prior to 1980 (*Rota et al., 1990*). Mutations to the HA1 region of the hemagglutinin protein are thought to drive the majority of antigenic drift in the influenza virus (*Wiley et al., 1981*; *Nelson and Holmes, 2007*). Experimental characterization of antigenic phenotype is possible through the hemagglutination inhibition (HI) assay (*Hirst, 1943*), which measures the cross-reactivity of one virus strain to serum raised against another strain through challenge or vaccination. Sera raised against older strains react poorly to more recent viruses resulting in new strains having a selective advantage over previously established strains.

The results of many HI assays across a multitude of viruses of a single subtype can be combined to yield a two-dimensional map, quantifying antigenic similarity and distance (*Smith et al., 2004*). The antigenic map of influenza A/H3N2 has shown substantial evolution of the influenza virus population since its emergence in 1968. Evolution of antigenic phenotype appears punctuated with episodes of more rapid innovation interspersed by periods of relative stasis, whereas genetic evolution appears more continuous (*Smith et al., 2004*), suggesting that a relatively small number of genetic changes or combinations of genetic changes may drive changes in antigenic phenotype (*Koel et al., 2013*). The process of antigenic drift results in the rapid turnover of the virus population, so that despite mutation, genetic diversity among contemporaneous viruses remains low. Such population turnover is supported by phylogenetic analysis that shows a characteristically 'spindly' tree with a single predominant trunk lineage and transitory side branches that persist for only 1–5 years (*Fitch et al., 1997*).

Previously, the antigenic and genetic patterns of influenza evolution have been analyzed essentially in isolation. An antigenic map is constructed from a panel of HI measurements, and a phylogenetic tree is constructed from sequence data. However, the opportunity for a combined approach exists as both the antigenic map and the phylogenetic tree often contain many of the same isolates. Here, we implement a flexible Bayesian approach to jointly characterize the antigenic and genetic evolution of the influenza virus population. We apply this approach to investigate the dynamics of A/H3N2, A/H1N1, B/Vic and B/Yam viruses, and, for the first time, present detailed reconstructions of the antigenic dynamics of all four circulating influenza lineages.

## Results and discussion

### Antigenic and evolutionary cartography

To assess patterns of antigenic evolution among influenza strains, we implemented a Bayesian probabilistic analog of multidimensional scaling (MDS), referred to here as BMDS (see 'Materials and methods'). In this model, viruses and sera are given $N$-dimensional locations, thus specifying an 'antigenic map', such that distances between viruses and sera in this space are inversely proportional to cross-reactivity. In the BMDS model, a map distance of one antigenic unit translates to an expectation of a twofold drop in HI titer between virus and sera. Maps that produce pairwise distances most congruent with the observed titers will have a high likelihood and will be favored by the BMDS model. We integrate over sources of uncertainty, such as antigenic locations, in a flexible Bayesian fashion. We apply this model to HI measurements of virus isolates against post-infection ferret antisera for influenza A/H3N2, A/H1N1, B/Vic and B/Yam.

We begin with Bayesian analogs of the models used by *Smith et al. (2004)*, in which viruses and sera are represented as $N$-dimensional locations as described in the 'Antigenic cartography' section of 'Materials and methods'. In this case, 'serum potencies' are fixed to the maximum titers exhibited by particular ferret sera and give the baseline expectation for titer when virus and serum are antigenically identical. Potency differs between serum isolates due to experimental noise (e.g., variation in serum concentration), but also due to differential ferret immune responses, causing some serum isolates to inhibit hemagglutination at generally higher titers than other isolates. In this model, virus and serum locations follow an uninformative diffuse normal prior. After comparing models of differing dimensions, *Smith et al. (2004)* arrive at a 2D model as the preferred model for their data. *Smith et al. (2004)* implement a form of MDS, seeking to optimize virus and serum locations such that the sum of squared errors between expected and observed titers is minimized (*Equation 3* of 'Materials and methods'). Here, in implementing BMDS, we provide a likelihood function for the probability of observing HI data given virus and serum locations (*Equation 8* of 'Materials and methods') and seek to estimate model parameters through Bayesian inference using Markov Chain Monte Carlo (MCMC). However, the basic antigenic model describing drop in HI titer as proportional to Euclidean distance between virus and serum locations is identical between these methods.

We test model performance by constructing training datasets representing 90% of the HI measurements for each of the four influenza lineages and test datasets representing the remaining 10% of the measurements for each lineage. By fitting the BMDS model to the training dataset, we are able to predict HI titers in the test dataset and compare these predicted titers to observed titers. We find that a two-dimensional model has better predictive power than models of lower or higher dimension in all four influenza lineages (models 1–5; *Table 1*). We find that this 2D model performs well, yielding an average absolute predictive error of between 0.78 and 0.91 $\log_2$ HI titers across influenza lineages (model 2; *Table 1*), in line with the results of *Smith et al. (2004)*. Consequently, we specify a two-dimensional model in all subsequent analyses. The finding of a low-dimensional map across influenza lineages extends previous studies in A/H3N2 (*Smith et al., 2004*) and remains an interesting and fundamental empirical observation.

Previous work on influenza antigenic and genetic evolution has shown that antigenic distance accumulates with increasing genetic distance (*Hay and Gregory, 2001*; *Smith et al., 2004*; *Russell et al., 2008*). Here, we examine pairwise relationships between viruses and observe a correlation between amino acid mutations and antigenic distance (*Figure 1*) and a similar correlation between phylogenetic distance, measured in years, and antigenic distance (*Figure 1*). Thus, genetic relationships between viruses provide some predictive power to estimate antigenic distances in the absence of HI data. However, the magnitudes of the coefficients of determination $R^2$ are low (*Figure 1*), suggesting that genetic relationships alone will not completely resolve antigenic distances.

**Table 1.** Average absolute prediction error of $\log_2$ HI titer for test data across models and datasets

| Model | Data | Dimen | Location prior | Serum potency | Virus avidity | A/H3N2 | A/H1N1 | B/Vic | B/Yam |
|---|---|---|---|---|---|---|---|---|---|
| | | | | | | Test error | | | |
| 1 | HI | 1D | Uninformed | Fixed | None | 1.35 | 0.94 | 0.90 | 1.08 |
| 2 | HI | 2D | Uninformed | Fixed | None | 0.91 | 0.78 | 0.82 | 0.90 |
| 3 | HI | 3D | Uninformed | Fixed | None | 0.93 | 0.80 | 0.85 | 0.92 |
| 4 | HI | 4D | Uninformed | Fixed | None | 0.98 | 0.84 | 0.90 | 0.97 |
| 5 | HI | 5D | Uninformed | Fixed | None | 1.04 | 0.89 | 0.98 | 1.04 |
| 6 | HI/year | 2D | Drift | Fixed | None | 0.91 | 0.75 | 0.77 | 0.83 |
| 7 | HI/year/seq | 2D | Diffusion/Drift | Fixed | None | 0.89 | 0.74 | 0.74 | 0.83 |
| 8 | HI/year/seq | 2D | Diffusion/Drift | Estimated | None | 0.77 | 0.73 | 0.66 | 0.75 |
| 9 | HI/year/seq | 2D | Diffusion/Drift | Fixed | Estimated | 0.80 | 0.72 | 0.69 | 0.75 |
| 10 | HI/year/seq | 2D | Diffusion/Drift | Estimated | Estimated | 0.76 | 0.71 | 0.64 | 0.72 |

Consequently, we seek to flexibly incorporate genetic data by modeling antigenic phenotype as an evolutionary diffusion (*Lemey et al., 2010*), wherein a virus's antigenic character state evolves along branches of the phylogenetic tree according to a Brownian motion process (see 'Materials and methods'). The phylogenetic diffusion process acts as a prior on virus locations, so that genetically similar viruses are expected to share similar antigenic locations. The antigenic diffusion process includes both systematic drift with time and covariance induced by phylogenetic proximity. We examine the effects of including only systematic drift (model 6; *Table 1*) and systematic drift plus phylogenetic diffusion (model 7; *Table 1*), finding a small increase in predictive accuracy of between 0.02 and 0.08 $\log_2$ HI titers when both processes are included. The systematic drift process informs virus and serum locations by dates of isolation and

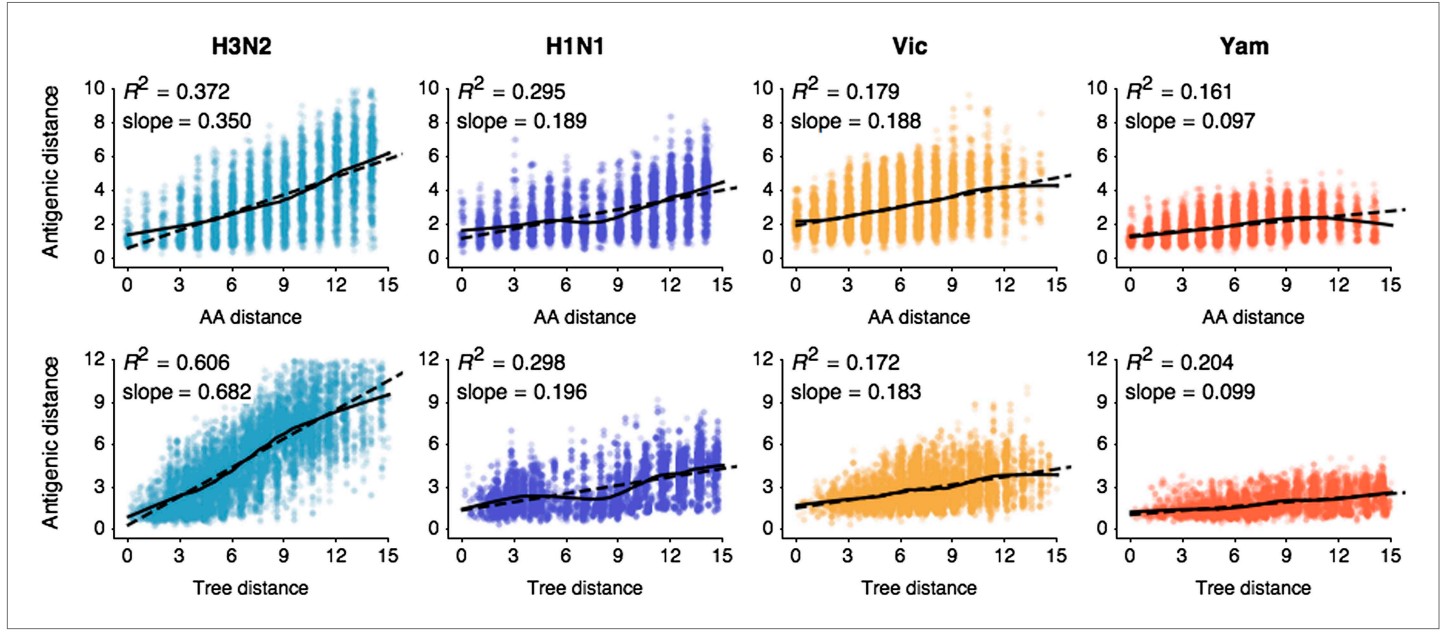

**Figure 1**. Pairwise correlations between genetic distance, measured as amino acid mutations or as phylogenetic distance, and antigenic distance for influenza A/H3N2, A/H1N1, B/Vic, and B/Yam. The top row shows correlations between number of amino acid mutations in HA1 and average antigenic distance between 10,000 random pairs of viruses. The bottom row shows correlations between average phylogenetic distance, measured in terms of years, and average antigenic distance between 10,000 random pairs of viruses. Dashed lines show linear model fits, with $R^2$ and slope noted, while solid lines show LOESS fits. Antigenic distances derive from model 2 of *Table 1*.

the phylogenetic diffusion process informs virus locations by genetic sequences. Thus, in these models, antigenic locations are inferred using both genetic data and HI data and will differ from locations inferred from HI data alone. If HI data is rich, then we expect minor differences in antigenic locations with the inclusion of genetic data (as may be the case for A/H3N2), while if HI data is spare, then we expect genetic data to play a larger role in determining antigenic locations (as may be the case for B/Vic and B/Yam).

We further extend the model by estimating serum potencies, rather than fixing serum potencies at maximum titers. Serum potencies differ across isolates due to experimental variation in serum extraction and processing or due to variation in ferret immune response. Serum potency determines the baseline expectation of titer when virus and serum have identical antigenic locations. However, if serum potency is fixed to the serum's maximum titer, this will often not be the case, as the virus giving the maximum titer may be antigenically distinct from the serum. Thus, fixing serum potencies will tend to under-estimate effect size; we observe a mean effect of 10.42 $\log_2$ HI titers for A/H3N2 when fixing serum potencies and a mean of 10.94 when estimating serum potencies. We find that estimating serum potencies improves test error further (model 8, *Table 1*), with improvements of between 0.01 and 0.12 $\log_2$ HI titers.

Additionally, we include and estimate 'virus avidity' in an analogous fashion, which is intended to represent differences in overall HI reactivity between viruses. Experimental work has demonstrated that influenza variants exist that differ in HA binding activity for cell surface glycan receptors, and that these high-avidity variants may arise in the presence of antibody pressure (*Hensley et al., 2009*). The presence of differential virus avidity has been previously shown to distort antigenic maps constructed from a model that disregards avidity effects (*Li et al., 2013*). Here, with virus avidities estimated, baseline titer derives from both the virus and the serum used in the HI reaction. We find that including virus avidities further improves test error, either with fixed serum potencies (model 9, *Table 1*) or with estimated serum potencies (model 10, *Table 1*). With fixed serum potencies, the inclusion of virus avidities results in improvements of between 0.02 and 0.09 $\log_2$ HI titers and with estimated serum potencies, the inclusion of virus avidities results in improvements of between 0.01 and 0.05 $\log_2$ HI titers.

We find that the average absolute error in predicted $\log_2$ HI titer is nearly constant with antigenic distance (Pearson correlation, $r = 0.098$), thus supporting our model assumption that the drop in $\log_2$ titer is proportional to the Euclidean distance separating viruses and sera on the antigenic map. Additionally, we find that the absolute error in predicted titer is nearly constant with time (Pearson correlation, $r = -0.007$). Antigenic locations inferred by the model are well resolved; estimates of antigenic distance between pairs of viruses show relatively little variation across the posterior. We estimate that virus distances have, on average, a 50% credible interval of ±0.45 antigenic units for A/H3N2, ±0.57 units for A/H1N1, ±0.76 units for B/Vic, and ±0.65 units for B/Yam.

We find strong correspondence between our results and previous results by *Smith et al. (2004)*, with equivalent models producing globally consistent antigenic maps and other models producing locally consistent maps with a small degree of global inconsistency (see 'Methods'). When implementing the same underlying model, differences in the MDS and BMDS approaches reflect greater philosophical differences between maximum-likelihood and Bayesian statistical approaches, with the former seeking the single most likely explanation for the data, and the latter seeking to fully characterize model uncertainty. Additionally, the BMDS method improves flexibility, allowing extensions to the basic cartographic model, such as the incorporation of virus avidities and evolutionary priors, that improve fit and add biological interpretability.

## Antigenic evolution across influenza lineages

Through our analysis, we reveal the antigenic, as well as evolutionary, relationships among viruses in influenza A/H3N2, A/H1N1, B/Vic and B/Yam, quantifying both antigenic and evolutionary distances between strains (*Figure 2*, *Figure 2—source data 1*). Over the time period of 1968 to 2011, influenza A/H3N2 shows substantially more antigenic evolution than is exhibited by A/H1N1 over the course of 1977 to 2009 or B/Vic and B/Yam over the course of 1986 to 2011. We observe prominent antigenic clusters in A/H3N2 and A/H1N1, but less prominent, though still apparent, clustering in B/Vic and B/Yam. Antigenic clusters show high genetic similarity, so that we observe very few mutation events leading to each cluster, rather than the repeated emergence of clusters. This analysis makes the fate of antigenic clusters obvious, with two clusters in A/H3N2 (Victoria/75 and Beijing/89) appearing to be evolutionary dead-ends. Labeling of prominent antigenic clusters in *Figure 2* is intended as a rough guide for orientation and not as exhaustive catalog of antigenic variation.

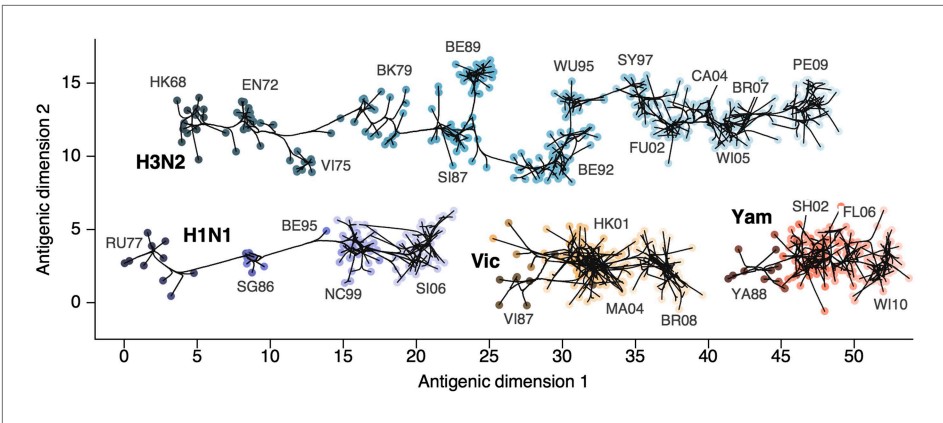

**Figure 2**. Antigenic locations of A/H3N2, A/H1N1, B/Vic, and B/Yam viruses showing evolutionary relationships between virus samples. Circles represent a posterior sample of virus locations and have been shaded based on year of isolation. Antigenic units represent twofold dilutions of the HI assay. Absolute positioning of lineages, for example A/H3N2 and A/H1N1, is arbitrary. Lines represent mean posterior diffusion paths when virus locations are fixed. Prominent antigenic clusters are labeled after vaccine strains present within clusters, and are abbreviated from Hong Kong/68, England/72, Victoria/75, Bangkok/79, Sichuan/87, Beijing/89, Beijing/92, Wuhan/95, Sydney/97, Fujian/02, California/04, Wisconsin/05, Brisbane/07, Perth/09 (A/H3N2), USSR/77, Singapore/86, Beijing/95, New Caledonia/99, Solomon Islands/06 (H1N1), Victoria/87, Hong Kong/01, Malaysia/04, Brisbane/08 (Vic), Yamagata/88, Shanghai/02, Florida/06, Wisconsin/10 (Yam).

The following source data are available for figure 2:

**Source data 1**. This tab-delimited text file lists every virus in *Figures 2 and 3*, including lineage, strain name, year of isolation, and coordinates in antigenic dimensions 1 and 2.

HI assays lack sensitivity beyond a certain point, so that for A/H3N2, cross-reactive measurements only exist between strains sampled at most 14 years apart, leaving only threshold titers, for example '<40', in more temporally distant comparisons. Because of the threshold of sensitivity of the HI assay, it is difficult to distinguish a linear trajectory in 2D antigenic space from a slightly curved trajectory (see 'Materials and Methods'). To solve this problem of identifiability, we assumed a weak prior that favors linear movement in the 2D antigenic space (present in models 6 through 9; *Table 1*), with the slope of the linear relationship and the precision of the relationship incorporated into the Bayesian model (see 'Materials and methods'). Because of this, we interpret map locations locally rather than globally, and assess rates of antigenic movement without making strong statements about the larger configuration under which the movement occurs.

We find that influenza A/H3N2 evolved along antigenic dimension 1 at an estimated rate of 1.01 antigenic units per year (*Figure 3*, *Figure 3—source data 1*; *Table 2*). However, we observe occasional large jumps in antigenic phenotype (*Figure 3*), corresponding to cluster transitions identified by *Smith et al. (2004)*. Most variation is contained within the first antigenic dimension, but dimension 2 occasionally shows variation when two antigenically distinct lineages emerge and transiently coexist (*Figure 2*), as is the case with the previously identified Beijing/89 and Beijing/92 clusters.

We find that other lineages of influenza evolved in antigenic phenotype substantially slower than A/H3N2 (*Figure 3*, *Table 2*). Influenza A/H1N1 evolved at a rate of 0.62 units per year, but showed a similar pattern of punctuated antigenic evolution with occasional larger jumps in phenotype, such as the emergence of the Solomon Islands/06 cluster. Influenza B/Victoria and B/Yamagata evolved slower still, with mean estimated rates 0.42 units per year and 0.32 units per year, respectively. Punctuated evolution is less obvious in B/Yam and B/Vic compared to A/H3N2 and A/H1N1, but antigenic clusters are still apparent, with recent transitions to the Brisbane/08 cluster in B/Vic (*Barr et al., 2010*) and to the Wisconsin/10 cluster in B/Yam (*Klimov et al., 2012*). Interestingly, a minor lineage of B/Vic, denoted B/Hubei-Songzi/51/2008 (*Barr et al., 2010*), has persisted through 2011, while remaining antigenically distinct from B/Brisbane/60/2008 viruses (*Figure 3*). Although we observe significantly different drift rates between lineages, we observe less variation in diffusion volatility (*Table 2*). This is

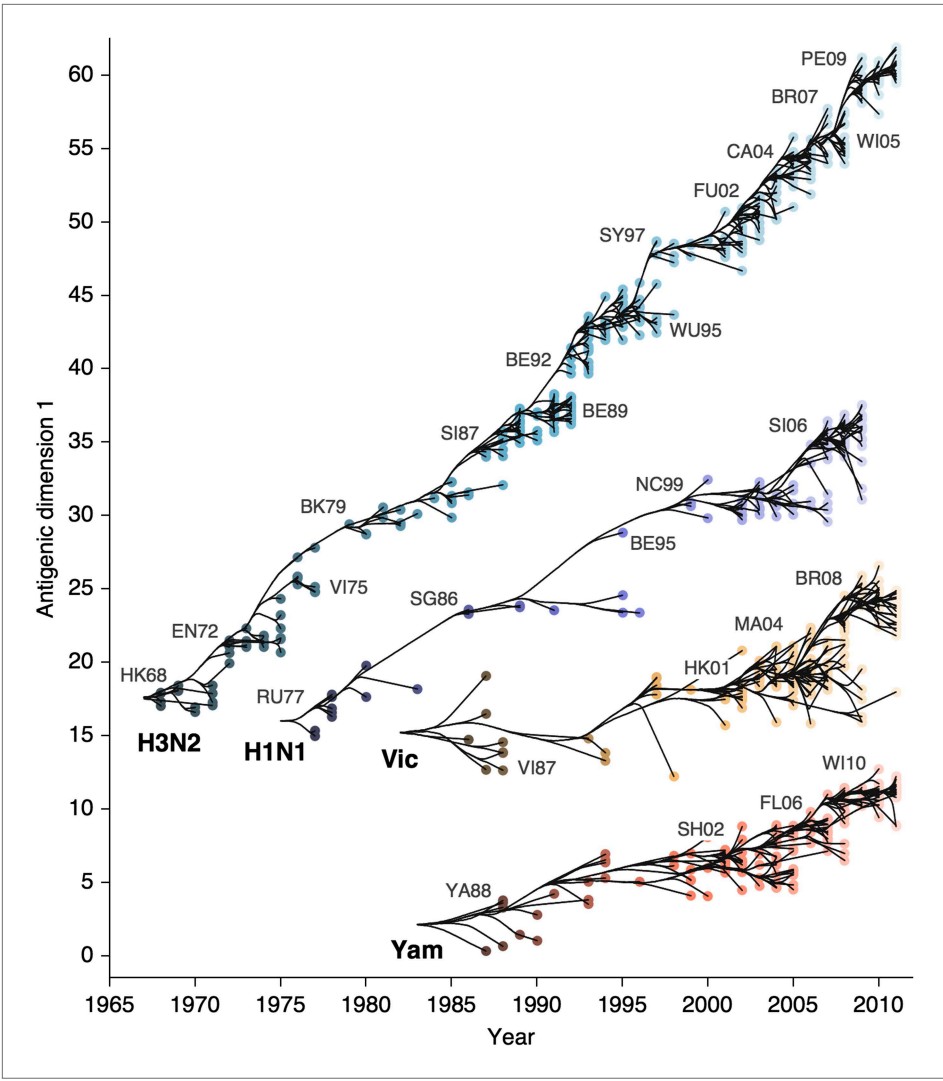

**Figure 3**. Antigenic drift of A/H3N2, A/H1N1, B/Vic and B/Yam viruses showing evolutionary relationships between virus samples. Antigenic drift is shown in terms of change of location in the first antigenic dimension through time. Circles represent a posterior sample of virus locations and have been shaded based on year of isolation. Antigenic units represent twofold dilutions of the HI assay. Relative positioning of lineages, for example A/H3N2 and A/H1N1, in the vertical axis is arbitrary. Lines represent mean posterior diffusion paths when virus locations are fixed. Prominent antigenic clusters are labeled after vaccine strains present within clusters, and are abbreviated from Hong Kong/68, England/72, Victoria/75, Bangkok/79, Sichuan/87, Beijing/89, Beijing/92, Wuhan/95, Sydney/97, Fujian/02, California/04, Wisconsin/05, Brisbane/07, Perth/09 (A/H3N2), USSR/77, Singapore/86, Beijing/95, New Caledonia/99, Solomon Islands/06 (H1N1), Victoria/87, Hong Kong/01, Malaysia/04, Brisbane/08 (Vic), Yamagata/88, Shanghai/02, Florida/06, Wisconsin/10 (Yam).

The following source data are available for figure 3:

**Source data 1**. This tab-delimited text file lists every virus in *Figures 2 and 3*, including lineage, strain name, year of isolation, and coordinates in antigenic dimensions 1 and 2.

---

reflected in *Figure 3*, where all four lineages exhibit similar levels of standing antigenic variation, despite A/H3N2 drifting more quickly in antigenic phenotype.

These patterns of antigenic drift influence the corresponding virus phylogenies (*Figure 4*). Influenza A/H3N2 has a characteristically spindly tree showing rapid turnover of the virus population, while A/H1N1 and B have trees that show greater degrees of viral coexistance (*Figure 4*). The scaled effective population size $N_e\tau$ measures the timescale of coalescence of a phylogeny and quantifies the visual

**Table 2.** Estimates of drift rate $\mu$ (in units per year), diffusion volatility $\sigma_x^2$ (in units$^2$ per year) and scaled effective population size $N_e\tau$ (in years) for influenza A/H3N2, A/H1N1, B/Vic and B/Yam including posterior means and 95% highest posterior density intervals

| Lineage | Drift $\mu$ | Volatility $\sigma_x^2$ | Effective pop size $N_e\tau$ |
|---|---|---|---|
| A/H3N2 | 1.01 (0.98–1.04) | 1.25 (0.98–2.35) | 5.03 (4.42–5.73) |
| A/H1N1 | 0.62 (0.56–0.67) | 0.92 (0.65–1.56) | 6.38 (4.99–8.12) |
| B/Vic | 0.42 (0.32–0.51) | 1.22 (0.85–2.25) | 10.40 (8.42–12.80) |
| B/Yam | 0.32 (0.25–0.39) | 0.71 (0.46–1.36) | 9.48 (7.76–11.50) |

distinction between a 'spindly' tree and a 'bushy' tree (*Bedford et al., 2011*). In this case, $N_e$ represents the number of concurrent infections in a panmictic population with generation interval $\tau$ (time separating infections up the genealogical tree), so that $N_e\tau$ is measured in terms of years and gives the expected waiting time for two randomly chosen lineages to coalesce in the genealogical tree. We see that $N_e\tau$ broadly correlates with the rate of antigenic drift (*Table 2*), with A/H3N2 showing fast drift and reduced effective population size as expected from basic epidemiological models (*Bedford et al., 2012*). Antigenic drift results in the replacement of antigenically primitive lineages by antigenically advanced lineages, thereby reducing genealogical diversity.

Thus, we observe a faster rate of antigenic drift in influenza A/H3N2 than in A/H1N1 or either lineage of influenza B. Previous work using general epidemiological models has suggested that rates of antigenic drift may be influenced by both the fundamental reproductive number $R_0$ and the rate at which

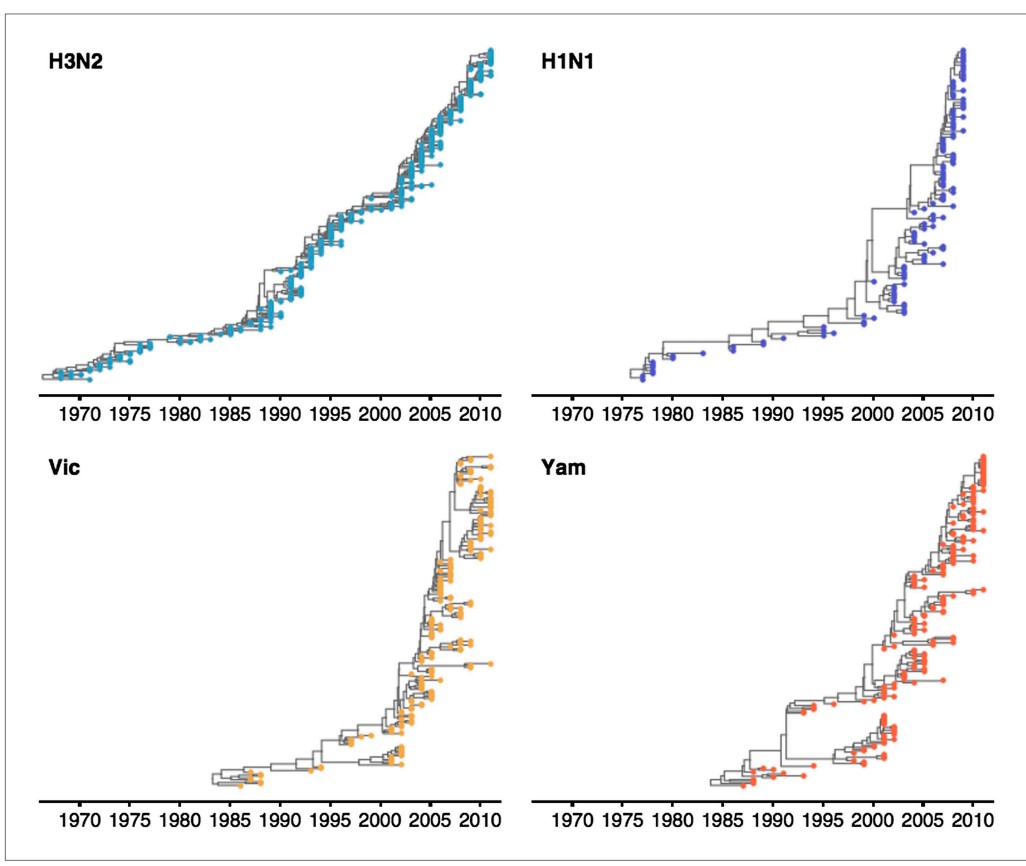

**Figure 4**. Time-resolved phylogenetic trees of A/H3N2, A/H1N1, B/Vic and B/Yam viruses. The maximum-clade credibility (MCC) tree is shown for each virus. These trees show genealogical relationships, so that branches are measured in terms of years rather than substitutions.

mutation decreases cross-immunity (*Gog and Grenfell, 2002*; *Lin et al., 2003*). Correspondingly, models specific to influenza evolution ascribe differences in the rate of antigenic drift of A/H3N2 relative to A/H1N1 and influenza B to either greater $R_0$ or greater mutation rate (*Ferguson et al., 2003*; *Bedford et al., 2012*). Without more detailed epidemiological modeling, the present study cannot conclusively distinguish between these causal possibilities.

## Punctuated evolution and its epidemiological consequences

We sought to summarize year-to-year patterns of antigenic drift by calculating the difference in mean virus location between consecutive years (*Figure 5A*). We estimate year-to-year antigenic drift for years 1992 to 2011 by calculating the average location along dimension 1 of phylogenetic lineages present in the tree at year $i$ and comparing this location to the average location of phylogenetic lineages present in the tree at year $i - 1$. There may often be large discontinuities in virus locations across the population; our use of difference in mean location is meant to capture both the distance between antigenic clusters and also the change in cluster frequency over consecutive years. We observe greater heterogeneity in year-to-year antigenic drift in type A than in type B lineages (*Figure 5B*), with standard deviation of year-to-year antigenic drift equal to 0.97 units in A/H3N2, 0.66 units in A/H1N1, 0.46 units in B/Vic and 0.26 units in B/Yam. This analysis classifies drift only to the level of consecutive years; some coarse-graining of the timings of transition events will necessarily occur.

We investigate the relationship between rates of antigenic drift and seasonal incidence in the USA in A/H3N2, A/H1N1, B/Vic and B/Yam. We measure seasonal incidence from USA CDC influenza surveillance reports for each virus lineage (A/H3N2, A/H1N1, B/Vic, B/Yam) by taking the average influenza-like (ILI) percentage in a season and multiplying this by the relative proportion of virus

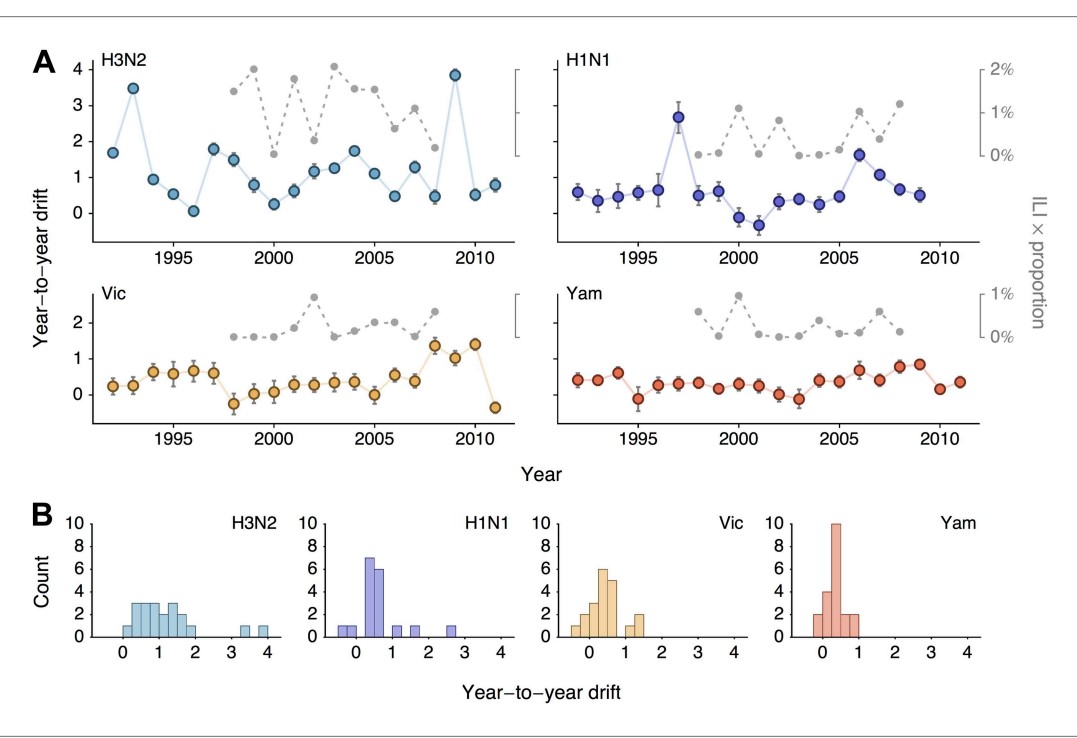

**Figure 5**. Year-to-year antigenic drift between 1992 and 2011 in A/H3N2, A/H1N1, B/Vic and B/Yam viruses. (**A**) Timeseries of year-to-year antigenic drift between 1992 and 2011 in A/H3N2, A/H1N1, B/Vic and B/Yam viruses. Colored lines represent year-to-year antigenic drift, where drift for year $i$ is measured as the mean of antigenic dimension 1 of phylogenetic lineages in year $i$ compared to the mean of antigenic dimension 1 of phylogenetic lineages from the previous year $i - 1$. For example, 2000 represents difference in antigenic dimension 1 between viruses from 1999 to 2000. Error bars represent 50% Bayesian credible intervals of year-to-year drift. Gray dotted lines represent lineage-specific seasonal incidence in the USA taken as average influenza-like illness (ILI) multiplied by proportion of viruses attributable to a lineage for each season. Here, 2000 represents the 2000/2001 influenza season. (**B**) Distribution of year-to-year antigenic drift between 1992 and 2011 in A/H3N2, A/H1N1, B/Vic and B/Yam viruses. DOI: 10.7554/eLife.01914.011

isolations attributable to a particular influenza lineage in a season (see 'Materials and methods'). This measure of incidence has previously been shown to have have predictive power in the analysis of seasonal influenza trends (*Goldstein et al., 2011*). We analyze incidence from the 1998/1999 to the 2008/2009 seasons to avoid possible complications from the 2009 pandemic. We begin by comparing overall rates of antigenic drift (*Figure 3*, *Table 2*) to overall levels of seasonal incidence across influenza lineages, finding a significant correlation between rate of antigenic drift and relative incidence across the four lineages (Pearson correlation, $r = 0.97$, $p = 0.041$).

We follow-up this analysis with a more detailed analysis of year-to-year variation in antigenic drift and lineage-specific incidence, comparing incidence in a season to antigenic drift of viruses coming into this season (*Figure 5A*). For example, we compare antigenic drift of viruses from 2000 to 2001 to incidence in the 2001/2002 season. Within each virus lineage, we find that years with pronounced antigenic drift tend to show increased incidence (*Figure 6*), finding Pearson correlation coefficients of 0.51, 0.29, 0.44 and 0.14 for A/H3N2, A/H1N1, B/Vic and B/Yam respectively. We calculated significance using bootstrap permutation tests finding *p*-values of 0.056, 0.201, 0.097, and 0.341 respectively. We applied a similar bootstrap permutation test to calculate the significance of finding the observed degree of correlation across all four lineages, arriving at a *p*-value of 0.018. The fact that we observe periods of pronounced antigenic drift preceding increased incidence in each of the four influenza lineages suggests a causal relationship, in which antigenic evolution drives increased incidence.

However, it is possible that if sampling count influences cartographic estimates then a spurious correlation could arise in which years with greater incidence have higher sample counts and artifactually high estimates of drift. We controlled for this possibility by testing to see if viral isolate count influences estimates of year-to-year drift by correlating drift between years $i$ and $i − 1$ against the ratio of the number of isolates from year $i$ to the number of isolates from year $i − 1$. We found little correlation when combining data across lineages (*Figure 7*, Pearson's $r = −0.01$). We tested significance following the same bootstrap procedure we used to assess the correlation between drift and incidence, finding a *p*-value of 0.514. Separating lineages gave *p*-values of 0.717, 0.246, 0.337, 0.504 for A/H3N2, A/H1N1, B/Vic and B/Yam, respectively. These findings suggest our results to be unbiased with regard to sample count.

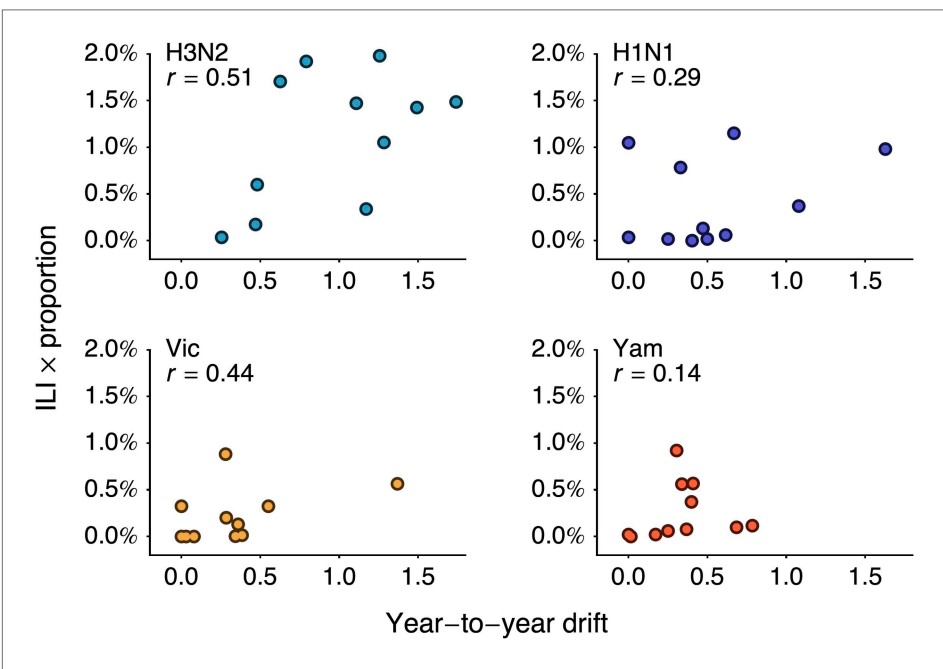

**Figure 6**. Relationship between antigenic drift and seasonal incidence for years 1998 to 2009 in influenza A/H3N2, A/H1N1, B/Vic and B/Yam. Antigenic drift from year $i − 1$ to year $i$ is compared to incidence in the season $i/i + 1$. For example, year-to-year antigenic drift from 2000 to 2001 is measured against incidence in the 2001/2002 season.

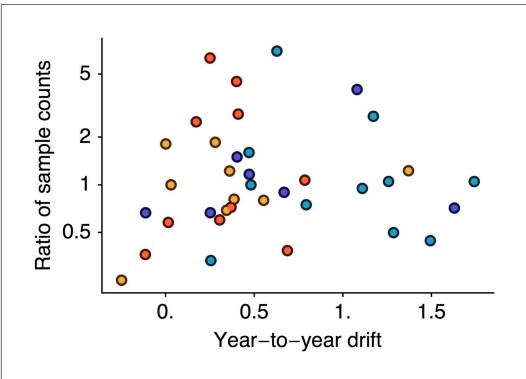

**Figure 7**. Relationship between antigenic drift and sample counts for years 1998 to 2011 in influenza A/H3N2, A/H1N1, B/Vic and B/Yam. Antigenic drift from year $i − 1$ to year $i$ is compared to the ratio of sample counts in year $i$ to counts in year $i − 1$. Only comparisons which had one or more samples in years $i − 1$ and $i$ were retained, leaving 11 A/H3N2, 7 A/H1N1, 9 B/Vic and 10 B/Yam comparisons. Points are colored according to influenza lineage based on the color scheme in *Figure 6*.

## Conclusions

Understanding antigenic evolution in seasonal influenza is crucial to our efforts of surveillance and control. Cartographic methods allow complex HI datasets to be compressed to more approachable location-based summaries that quantify antigenic relationships between strains, including relationships not directly measured via HI. In this study, we provide a foundation for evolutionary antigenic cartography, which seeks to simultaneously assess antigenic phenotype and antigenic evolution. We use this approach to characterize competitive dynamics across influenza lineages A/H3N2, A/H1N1, B/Vic and B/Yam and show that antigenic evolution within each lineage drives strain replacement and contributes to seasonal incidence patterns. We find that influenza A/H3N2 evolves faster in antigenic phenotype than A/H1N1, which in turn evolves faster than B/Vic or B/Yam. Consequently, the influenza A/H3N2 virus population turns over more quickly than A/H1N1 or influenza B, exhibiting a smaller effective population size and a 'spindlier' phylogenetic tree. Furthermore, we observe a correlation between antigenic drift and viral incidence both across and within influenza lineages. The finding that antigenic evolution correlates with subsequent increased incidence within a lineage suggests a causal role for antigenic drift driving influenza incidence patterns. The correlation between incidence and drift further suggests the possibility of using HI data at the start of an influenza season to predict which lineage will subsequently predominate.

The statistical framework presented here represents a baseline to which further advancements in modeling antigenic phenotype and evolution may be made. For example, our likelihood-based model facilitates the inclusion of possible covariates affecting immunological titer, which could include experimental factors such as red blood cell type used in the HI assay (*Lin et al., 2012*) and whether oseltamivir is included in the HI reaction (*Lin et al., 2010*). Additionally, this framework should be ideally suited to uncovering genetic determinants of antigenic change, as both the sequence state and antigenic location of internal nodes in the phylogeny may be estimated. In this fashion, it should be possible to correlate sequence substitutions directly to antigenic diffusion. Identifying viruses that will come to predominate in the global virus population while they are still at low frequency remains an enormous challenge. However, combining evolutionary and antigenic information may eventually prove useful in identifying low-frequency, but expanding, lineages of antigenically novel viruses that represent ideal targets for vaccine strain selection.

## Materials and methods

### Antigenic cartography

Antigenic characteristics of viral strains are often assessed through immunological assays such as the hemagglutination inhibition (HI) assay (*Hirst, 1943*). At heart, these assays compare the reactivity of one virus strain to antibodies raised against another virus strain via challenge or vaccination. In the case of HI, the measurement of cross-reactivity takes the form of a titer representing the dilution factor at which serum raised against a particular virus ceases to be effective at inhibiting the binding of another virus to red blood cells. These factors are commonly assessed by serial dilution, so that HI titers will form a log series, 40, 80, 160, etc.... Because experimental HI titers typically differ by factors of two, we find it convenient to work in $\log_2$ space and represent the titer of virus $i$ against serum $j$ as $H_{ij} = \log_2$(HI titer), that is a titer of 160 has $H_{ij} = 7.32$. Due to experimental constraints, most comparisons cannot be made, leading to a sparse observation matrix $\mathbf{H} = \{H_{ij}\}$. Further, measurements are usually interval and truncated, for example inhibition may cease somewhere between the serial titers of 160 and 320, or inhibition may be absent at all titers assayed, suggesting a threshold somewhere between 0 and 40.

Previous work (**Smith et al., 2004**; **Cai et al., 2010**) has used multidimensional scaling (MDS) to place viruses and sera on an 'antigenic map'. These methods heuristically optimize locations of viruses and sera by seeking to minimize the sum of squared errors between titers predicted by map locations and observed titers. Antigenic maps produced by these methods have proven useful in categorizing virus phenotypes (**Smith et al., 2004**), but the extension of these methods to integrate genetic data remains notably lacking.

Here, we follow previous models in representing antigenic locations as points in a low $P$-dimensional antigenic map. One of our initial goals is to find an optimal projection of the high-dimensional distance matrix $\mathbf{H}$ into this lower dimensional space. We conduct this projection using Bayesian multidimensional scaling (BMDS) (**Oh and Raftery, 2001**) in which we construct a probabilistic model to quantify the fit of a particular configuration of cartographic locations to the observed matrix of serological measurements. Typically, $P = 2$, but higher or lower dimensions may better reflect the data.

Let $\mathbf{x}_i \in \mathbb{R}^P$ represent the cartographic location of virus $i$ for $i = 1,\ldots,$ n, so that $\mathbf{x}_i = (x_{i1}, x_{i2})'$ for $P = 2$. Similarly, let $\mathbf{y}_j$ represent the cartographic location of serum $j$ for $j = 1,\ldots, k$, so that $\mathbf{y}_j = (y_{j1}, y_{j2})'$ for $P = 2$. For notational compactness, we collect together all virus coordinates into an $n \times P$ matrix $\mathbf{X} = (\mathbf{x}_1,\ldots,\mathbf{x}_n)'$ and all serum coordinates into an $k \times P$ matrix $\mathbf{Y} = (\mathbf{y}_1,\ldots,\mathbf{y}_k)'$. Virus and serum may be isolated from/raised against the same strain and have different cartographic locations, and separate serum isolates raised against the same strain may also have different cartographic locations. This gives a set of distances between virus and serum cartographic locations.

$$\delta_{ij} = \| \mathbf{x}_i - \mathbf{y}_j \|_2, \tag{1}$$

where, $\| \cdot \|_2$ is an $L_2$ norm.

Traditional approaches to antigenic cartography (**Smith et al., 2004**) begin by defining immunological distance as

$$d_{ij} = s_j - H_{ij}, \tag{2}$$

where, $H_{ij}$ is the $\log_2$ titer of virus $i$ against serum $j$ and serum potency $s_j = \max(H_{1j},\ldots,H_{nj})$ is fixed. In following multidimensional scaling, these approaches attempt to optimize over unknown $\mathbf{X}$ and $\mathbf{Y}$ such that

$$\sum_{(i,j)\in\mathcal{I}} \left( \delta_{ij} - d_{ij} \right)^2 \tag{3}$$

is minimized, where, $\mathcal{I} = \{(i,j): H_{ij} \text{ is measured}\}$. In the case of threshold measurements, this error function is modified slightly; see **Smith et al. (2004)** for further details.

Here, we instead assume a probabilistic interpretation in which an observed titer is normally distributed around its cartographic expectation with variance $\varphi^2$,

$$H_{ij} \sim \mathcal{N}\left(s_j - \delta_{ij}, \varphi^2\right). \tag{4}$$

Consequently, the likelihood of observing an exact titer given the placement of antigenic locations is

$$f_|\left(H_{ij}\right) = \phi\left(\frac{H_{ij} + \delta_{ij} - s_j}{\varphi}\right), \tag{5}$$

where, $\phi(\cdot)$ represents the standard normal probability density function (PDF). Previous BMDS has employed a sampling density truncated to strictly positive quantities since $d_{ij}$ are directly observed, non-negative quantities. In the antigenic setting, these remain random and can be negative since neither $s_j$ is known nor is $H_{ij}$ observed with much precision.

HI assays sometimes show no inhibition at all measured titrations, for example a measurement can be reported as '<40'. In this case, the likelihood of observing the threshold measurement follows the cumulative density of the lower tail of the normal distribution.

$$f_\lrcorner\left(H_{ij}\right) = \Phi\left(\frac{H_{ij} + \delta_{ij} - s_j}{\varphi}\right), \tag{6}$$

where, $\Phi(\cdot)$ represents the standard normal cumulative distribution function (CDF). Although it is simplest to assume that immunological measurements represent point estimates, it seems more natural to assume that the threshold for inhibition occurs between two titers, for example we observe inhibition at 1:160 dilution and no inhibition at 1:320 dilution. Rather than taking the HI titer as 160, we can instead treat this as an interval measurement, assuming that the exact titer for inhibition would occur somewhere

between 160 and 320. HI titers are usually reported as the highest titer that successfully inhibits virus binding, so that in this case, we calculate the likelihood of an interval measurement as

$$f_{\sqcup}\left(H_{ij}\right) = \Phi\left(\frac{H_{ij} + \delta_{ij} - s_j + 1}{\varphi}\right) - \Phi\left(\frac{H_{ij} + \delta_{ij} - s_j}{\varphi}\right). \tag{7}$$

These likelihoods are illustrated in *Figure 8*. Throughout our analyses, we use interval likelihoods $f_{\sqcup}$ rather than point likelihoods $f_{|}$ unless otherwise noted.

We calculate the overall likelihood by multiplying probabilities of individual measurements

$$L(\mathbf{X},\mathbf{Y}) = \prod_{(i,j)\in\mathcal{I}} f\left(H_{ij}\right), \tag{8}$$

using probability functions $f_{|}$, $f_{\lrcorner}$ and $f_{\sqcup}$ as appropriate. We begin by assuming independent, diffuse normal priors on virus and serum locations.

$$\begin{aligned}\mathbf{x}_i &\sim \mathcal{N}\left(\mathbf{m},\Sigma\right)\\\mathbf{y}_j &\sim \mathcal{N}\left(\mathbf{m},\Sigma\right),\end{aligned} \tag{9}$$

where, $\mathbf{m} = (0,\ldots,0)'$ and $\Sigma$ is a diagonal matrix with diagonal elements all equal to 10,000.

## Virus avidity and serum potency

The preceding model represents immunological distance as a drop in titer against the most reactive comparison for a particular serum. However, this model may be biased in some circumstances. In one example, if a particular serum $j$ is only measured against distant viruses, its maximum titer will be artificially low and the likelihoods concerning this serum will appear poor. To address this issue, we relax the assumption of fixed $s_j$ values and treat the expected $\log_2$ titer when $\delta_{ij} = 0$ as a random variable. In this case, $H_{ij}$ still follows *Equation 4* with expectation $s_j - \delta_{ij}$, but the vector of 'serum potencies' $s = (s_1,\ldots, s_k)$ is random and estimated rather than fixed. We assume that $s_j$ values are hierarchically distributed according to a normal distribution. We take an Empirical Bayesian approach in specifying the mean

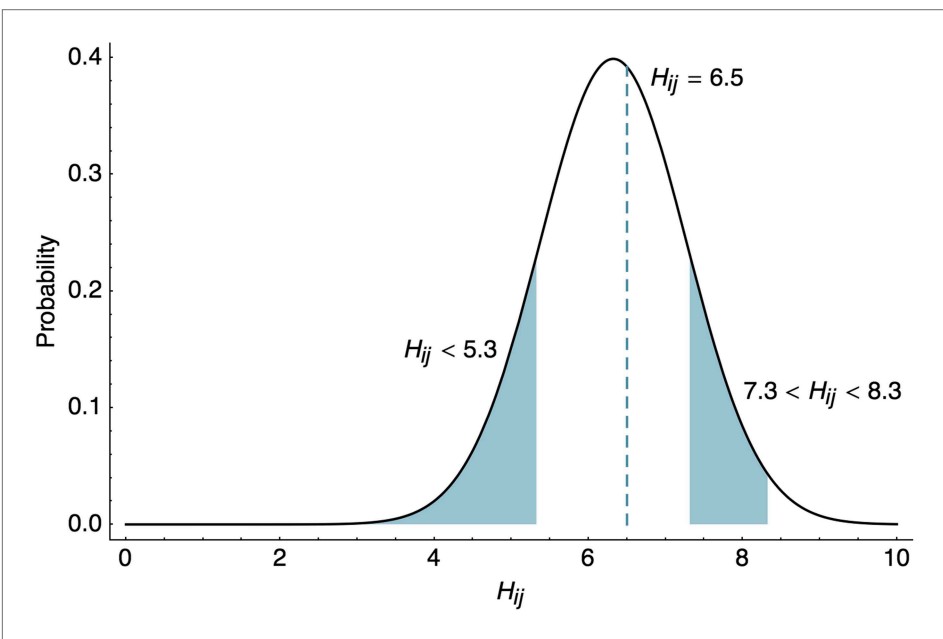

**Figure 8**. Likelihood of HI titers in the BMDS model. Here, we show the likelihoods of observing three different outcomes given $\delta_{ij} = 4$, $\varphi = 0.95$, and $s_j = \log_2(1280) = 10.32$. The likelihood of observing a threshold titer of '<40' is equal to the lower tail of the probability density function $f_{\lrcorner}(5.32) = 0.146$. The likelihood of observing a point measurement with an exact inhibiting titer of '90.5' is equal to the density function $f_{|}(6.5) = 0.413$. The likelihood of observing an interval measurement with an inhibiting titer somewhere between '160' and '320' is equal to $f_{\sqcup}(7.32) = 0.129$.

and variance of this distribution, set to the empirical mean and empirical variance of the set of maximum titers across sera $\{\max(H_{1j},\ldots,H_{nj}) : j = 1,\ldots,k\}$. This formulation assumes that particular sera are more reactive in general than other sera.

Additionally, we follow the same logic and assume that some virus isolates are more reactive than other virus isolates and include a parameter for 'virus avidity' $v_i$ representing the general level of reactivity across HI assays. With virus avidity included, observed titers follow

$$H_{ij} \sim \mathcal{N}\left(\frac{v_i + s_j}{2} - \delta_{ij}, \varphi^2\right), \tag{10}$$

and the vector of virus avidities $v_i$ for $i = 1,\ldots,n$ is estimated in an analogous hierarchical fashion, with $\mathbf{v}$ normally distributed with mean and variance equal to the empirical mean and variance of the set of maximum titers across viruses $\{\max(H_{i1},\ldots,H_{ik}) : i = 1,\ldots,n\}$.

## Drift model of antigenic evolution

As presented, multiple configurations of virus and serum locations $\mathbf{X}$ and $\mathbf{Y}$ will give the same likelihood of an observed data matrix $\mathbf{H}$. An example of this phenomenon is shown in *Figure 9*. In this case, it is impossible to determine from the HI data at hand whether the blue and yellow viruses are antigenically similar (*Figure 9A*) or antigenically divergent (*Figure 9B*). This presents an issue of model identifiability, where absolute, as opposed to relative, antigenic locations cannot be determined from observing the serological data alone. Thus, in order to achieve a more interpretable model we impose a weak prior on global locations. In influenza, it's clear that antigenic distance between strains increases with time (*Smith et al., 2004*; *Cai et al., 2010*). To capture this, we replace our previous diffuse prior with an informed prior in which the expected location of viruses and sera increases with date of sampling along dimension one, and each virus and serum location follows an independent normal distribution centered around this temporal expectation, so that

$$x_{i1} \sim \mu t_i + \mathcal{N}\left(0, \sigma_x^2\right)$$
$$y_{j1} \sim \mu t_j + \mathcal{N}\left(0, \sigma_y^2\right), \tag{11}$$

where, $t$ is the difference between the date of the indexed virus or serum and the date of the earliest sampled virus or serum, and other dimensions follow $x_{im} \sim \mathcal{N}\left(0, \sigma_x^2\right)$ and $y_{jm} \sim \mathcal{N}\left(0, \sigma_y^2\right)$ for $m \geq 2$. Thus,

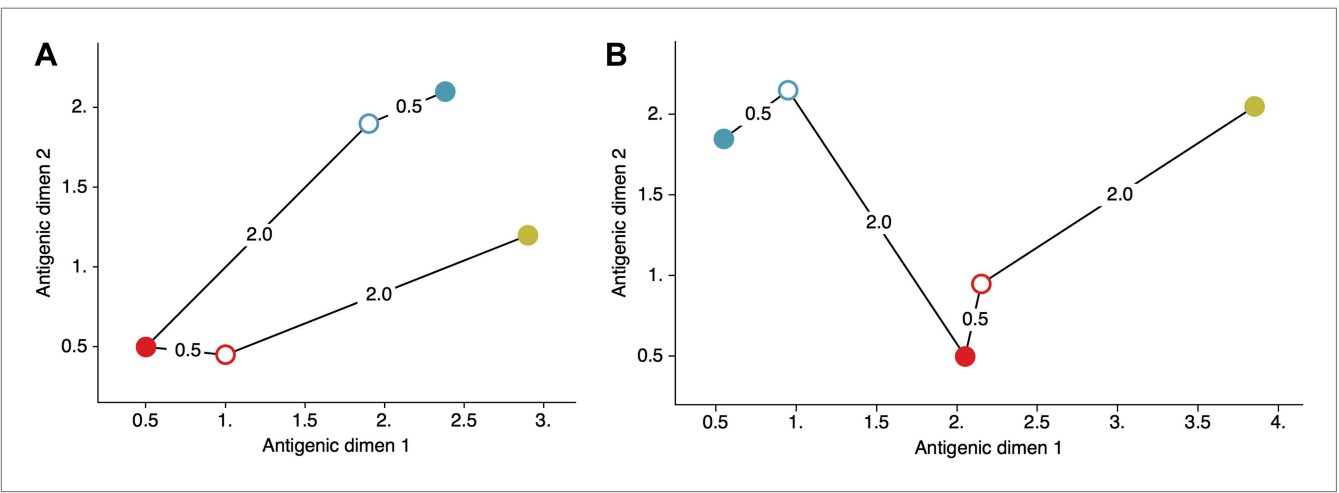

**Figure 9**. Schematic antigenic map with three viruses and two sera. (**A**) Map with virus 1 and virus 3 antigenically similar. (**B**) Map with virus 1 and virus 3 antigenically divergent. Virus 1 is shown in blue, virus 2 is shown in red and virus 3 is shown in yellow. Virus isolates are represented by filled circles, sera raised against viruses are shown as open circles and map distances $\delta_{ij}$ are shown as solid lines connecting viruses and sera. Sera from virus 1 is compared against viruses 1 and 2, while sera from virus 2 is compared against viruses 2 and 3. Configurations (**A**) and (**B**) represent cartographic models that would give equal likelihoods to a set of serological data $\{H_{11}, H_{21}, H_{22}, H_{32}\}$.

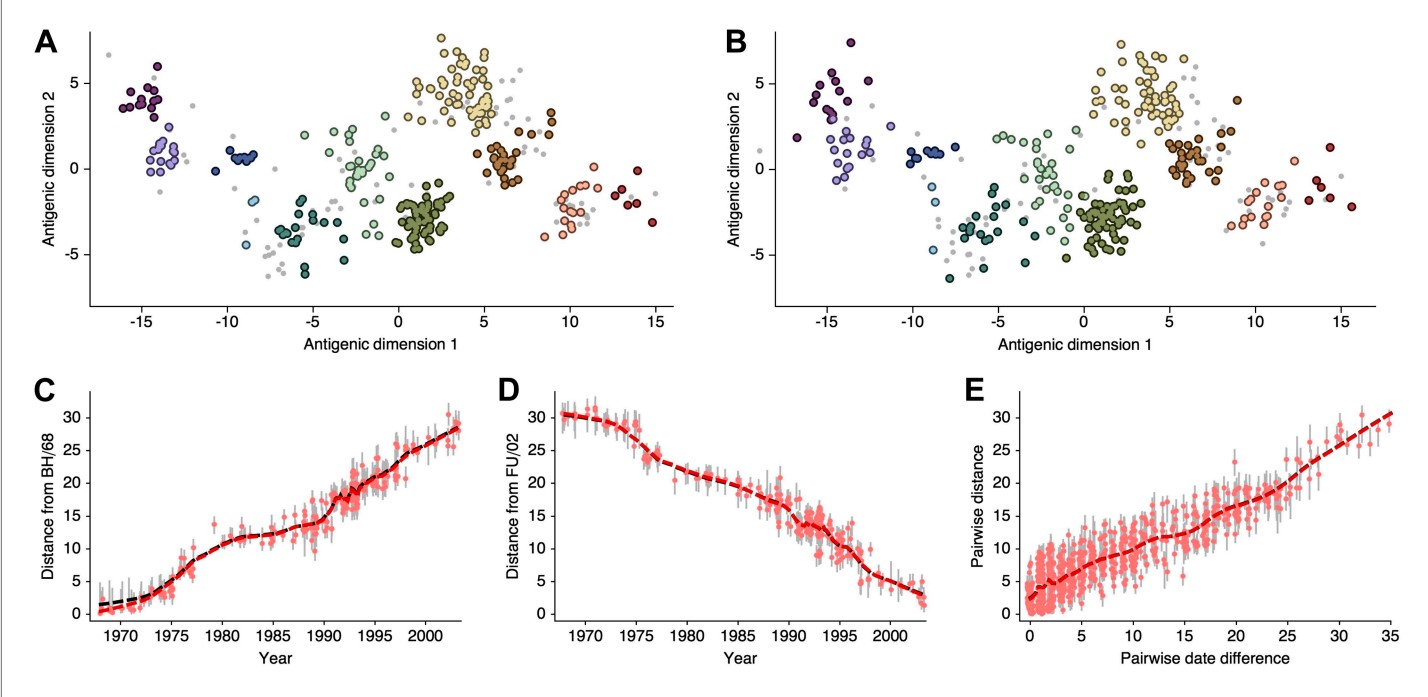

**Figure 10**. Comparison of A/H3N2 antigenic locations estimated by **Smith et al. (2004)** using MDS and an equivalent BMDS model. (**A**) MDS antigenic locations, reoriented so that the primary dimension lies on the *x*-axis rather than on the *y*-axis as in Figure 1 of **Smith et al. (2004)**. (**B**) A posterior sample of antigenic locations from an equivalent BMDS model. In (**A**) and (**B**), viruses are shown as colored circles, with color denoting antigenic cluster inferred by (**Smith et al., 2004**), and sera are shown as gray points. (**C**) Antigenic distances between A/Bilthoven/16,190/1968 and all other viruses determined for both methods. (**D**) Antigenic distances between A/Fujian/411/2002 and all other viruses determined for both methods. (**E**) Antigenic distances between 750 random pairs of viruses determined for both methods. In (**C**), (**D**) and (**E**) red points show distances for the MDS model and gray bars show the 95% credible interval of distances for the BMDS model, while the red dashed line shows a LOESS regression to MDS distances, and the black dashed line shows a LOESS regression to the BMDS distances. The BMDS model has a Uniform (−100, 100) prior on antigenic locations and serum potencies fixed at maximum titer values.

The following source data are available for figure 10:

**Source data 1**. This tab-delimited text file lists every virus and serum in **Figure 10**, including strain name, year of isolation, coordinates in antigenic dimensions 1 and 2, and potency for sera.

this model assumes that virus and serum locations drift in a line across the antigenic map at rate $\mu$. The parameter $\sigma_x$ determines the breadth of the cloud of virus locations at each point in time, while $\sigma_y$ determines the breadth of the cloud of serum locations.

## Phylogenetic diffusion model of antigenic evolution

We simultaneously model antigenic locations and genetic relatedness by assuming that virus locations are influenced by evolution following a Brownian motion process (**Lemey et al., 2010**). To do this, we replace the previous prior specifying independent virus locations with a prior that incorporates covariance based on shared evolutionary history.

$$\mathbf{X} \sim \begin{pmatrix} \mu t_1 & 0 \\ \vdots & \vdots \\ \mu t_n & 0 \end{pmatrix} + \text{Evolutionary Brownian Process}\left(\sigma_x, \tau\right) \quad (12)$$

for $P = 2$, where, $\sigma_x$ is the volatility parameter of the Brownian motion over virus locations and $\tau$ is a phylogeny specifying tree topology and branch lengths. Thus, viruses that are genetically similar are induced to have prior locations close to one another on the antigenic map. In the evolutionary Brownian process, the tips of the phylogeny $\tau$ correspond to the set of virus locations ($\mathbf{x}_1, \ldots, \mathbf{x}_n$), and the probability of observing tip locations depends on the locations of internal nodes ($\mathbf{x}_{n+1}, \ldots, \mathbf{x}_{2n-2}$) and on the location

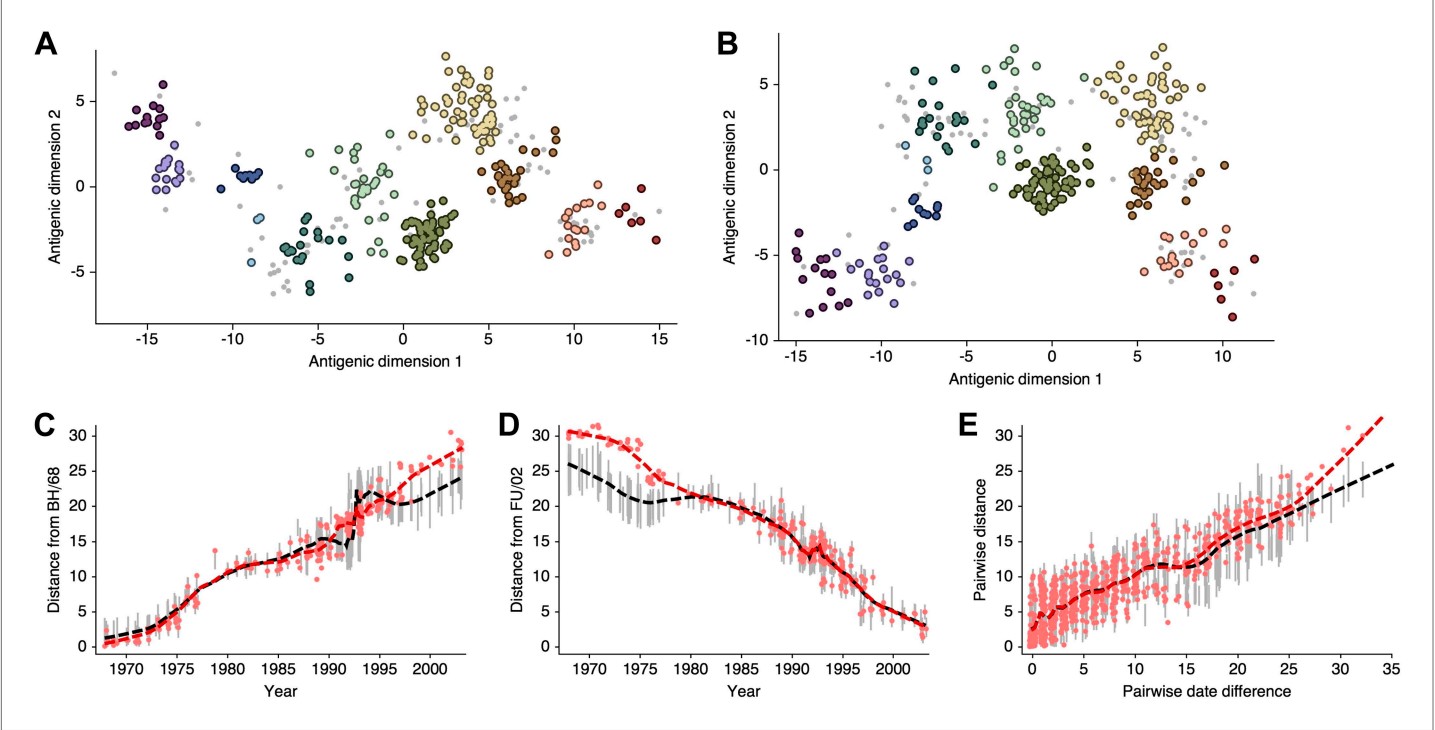

**Figure 11**. Comparison of A/H3N2 antigenic locations estimated by *Smith et al. (2004)* using MDS and an equivalent BMDS model under an alternative solution. (**A**) MDS antigenic locations, reoriented so that the primary dimension lies on the *x*-axis rather than on the *y*-axis as in Figure 1 of *Smith et al. (2004)*. (**B**) A posterior sample of antigenic locations from an equivalent BMDS model that has converged on the alternative solution. In (**A**) and (**B**), viruses are shown as colored circles, with color denoting antigenic cluster inferred by *Smith et al. (2004)*, and sera are shown as gray points. (**C**) Antigenic distances between A/Bilthoven/16,190/1968 and all other viruses determined for both methods. (**D**) Antigenic distances between A/Fujian/411/2002 and all other viruses determined for both methods. (**E**) Antigenic distances between 750 random pairs of viruses determined for both methods. In (**C**), (**D**) and (**E**) red points show distances for the MDS model and gray bars show the 95% credible interval of distances for the BMDS model, while the red dashed line shows a LOESS regression to MDS distances, and the black dashed line shows a LOESS regression to the BMDS distances. The BMDS model has a Uniform (−100, 100) prior on antigenic locations and serum potencies fixed at maximum titer values.

The following source data are available for figure 11:

**Source data 1**. This tab-delimited text file lists every virus and serum in *Figure 11*, including strain name, year of isolation, coordinates in antigenic dimensions 1 and 2, and potency for sera.

of the root node $\mathbf{x}_{2n-2}$. This process assumes that a virus location $\mathbf{x}_i$ follows from the location of its parent virus $\mathbf{x}_{f(i)}$, and with the addition of drift along dimension 1, is distributed as

$$\mathbf{x}_i \sim \left(\mu d_i, 0\right)' + \mathcal{N}\left(\mathbf{x}_{f(i)}, d_i \Sigma\right) \tag{13}$$

for $P = 2$, where, $f(i)$ is a function that maps nodes to parental nodes, $d_i$ is the length of the branch connecting virus $i$ to parent virus $f(i)$, and $\Sigma$ is a diagonal matrix with diagonal elements all equal to $\sigma_x^2$. The root virus location $\mathbf{x}_{2n-1}$ is assumed to follow a normal distribution with expectation $\left(\mu t_{2n-1}, 0\right)'$ for $P = 2$ and variance determined by the diffusion volatility $\sigma_x$ (*Lemey et al., 2010*). The probability of virus locations $p\left(\mathbf{X} \mid \mu, \sigma_x, \tau\right)$ is determined through analytical integration across internal states following the methods introduced in (*Lemey et al., 2010*). This formulation corresponds to a Wiener process with drift, in which the drift term $\mu$ only influences the expected states of nodes along the phylogeny, but does not influence the covariance structure among these nodes, which remains the same as it does in a standard Wiener process (*Borodin and Salminen, 2002*). This allows the separation in *Equation 12* between drift terms affecting only expectations and the evolutionary Brownian process that includes covariance among virus locations $(\mathbf{x}_1, \dots, \mathbf{x}_n)$.

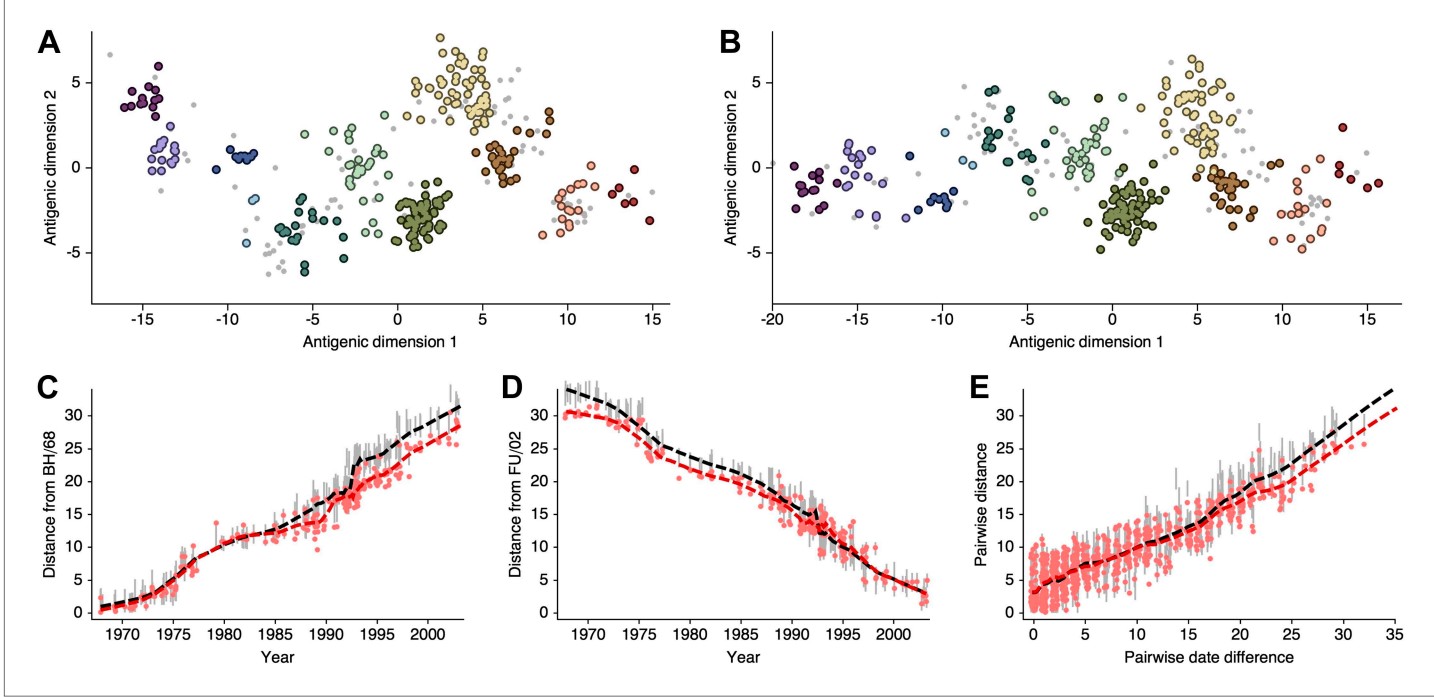

**Figure 12**. Comparison of A/H3N2 antigenic locations estimated by *Smith et al. (2004)* using MDS and an extended BMDS model that includes date-informed priors on antigenic locations. (**A**) MDS antigenic locations, reoriented so that the primary dimension lies on the x-axis rather than on the y-axis as in Figure 1 of *Smith et al. (2004)*. (**B**) A posterior sample of antigenic locations from a BMDS model that includes date-informed priors on antigenic locations. In (**A**) and (**B**), viruses are shown as colored circles, with color denoting antigenic cluster inferred by *Smith et al. (2004)*, and sera are shown as gray points. (**C**) Antigenic distances between A/Bilthoven/16,190/1968 and all other viruses determined for both methods. (**D**) Antigenic distances between A/Fujian/411/2002 and all other viruses determined for both methods. (**E**) Antigenic distances between 750 random pairs of viruses determined for both methods. In (**C**), (**D**) and (**E**) red points show distances for the MDS model and gray bars show the 95% credible interval of distances for the BMDS model, while the red dashed line shows a LOESS regression to MDS distances, and the black dashed line shows a LOESS regression to the BMDS distances. The BMDS model has a date-informed prior on antigenic locations and serum potencies fixed at maximum titer values.

The following source data are available for figure 12:

**Source data 1**. This tab-delimited text file lists every virus and serum in *Figure 12*, including strain name, year of isolation, coordinates in antigenic dimensions 1 and 2, and potency for sera.

In this study, the phylogenetic tree $\tau$ is estimated using sequence data for viruses 1,…, n according to well-established methods implemented in the software package BEAST (*Drummond et al., 2012*).

## Posterior inference

Top-level priors for $1/\varphi^2$, $\mu$, $1/\sigma_x^2$, and $1/\sigma_y^2$ are assumed to follow diffuse Gamma(a, b) distributions with a = 0.001 and b = 0.001. These diffuse priors were chosen to be non-informative and provide little-to-no weight on the resulting posterior distributions. Under the full model, the posterior probability of observing virus and serum locations given immunological data is factored.

$$p(\mathbf{X},\mathbf{Y}\,|\,\mathbf{H}) \propto p(\mathbf{H}\,|\,\mathbf{X},\mathbf{Y},\mathbf{s},\mathbf{v},\varphi)\,p(\mathbf{X}\,|\,\mu,\sigma_x,\tau)\,p(\mathbf{Y}\,|\,\mu,\sigma_y)\,p(\mathbf{s},\mathbf{v},\varphi,\mu,\sigma_x,\sigma_y,\tau). \qquad (14)$$

We sample from this posterior distribution using the MCMC procedures implemented in the software package BEAST (*Drummond et al., 2012*). Metropolis–Hastings proposals include transition kernels that translate individual virus and serum locations $\mathbf{x}_i$ and $\mathbf{y}_j$ and individual virus avidities $v_i$ and serum potencies $s_j$, and other transition kernels that scale the entire set of virus and serum locations $\mathbf{X}$ and $\mathbf{Y}$ and that scale parameters $\varphi$, $\mu$, $\sigma_x$ and $\sigma_y$. For the present analysis, a two-step approach was taken to sample phylogenies, where a posterior sample of phylogenies was gathered using sequence data and then, in the cartographic analysis, trees from this set were randomly proposed and accepted following the Metropolis–Hastings algorithm (*Pagel et al., 2004*).

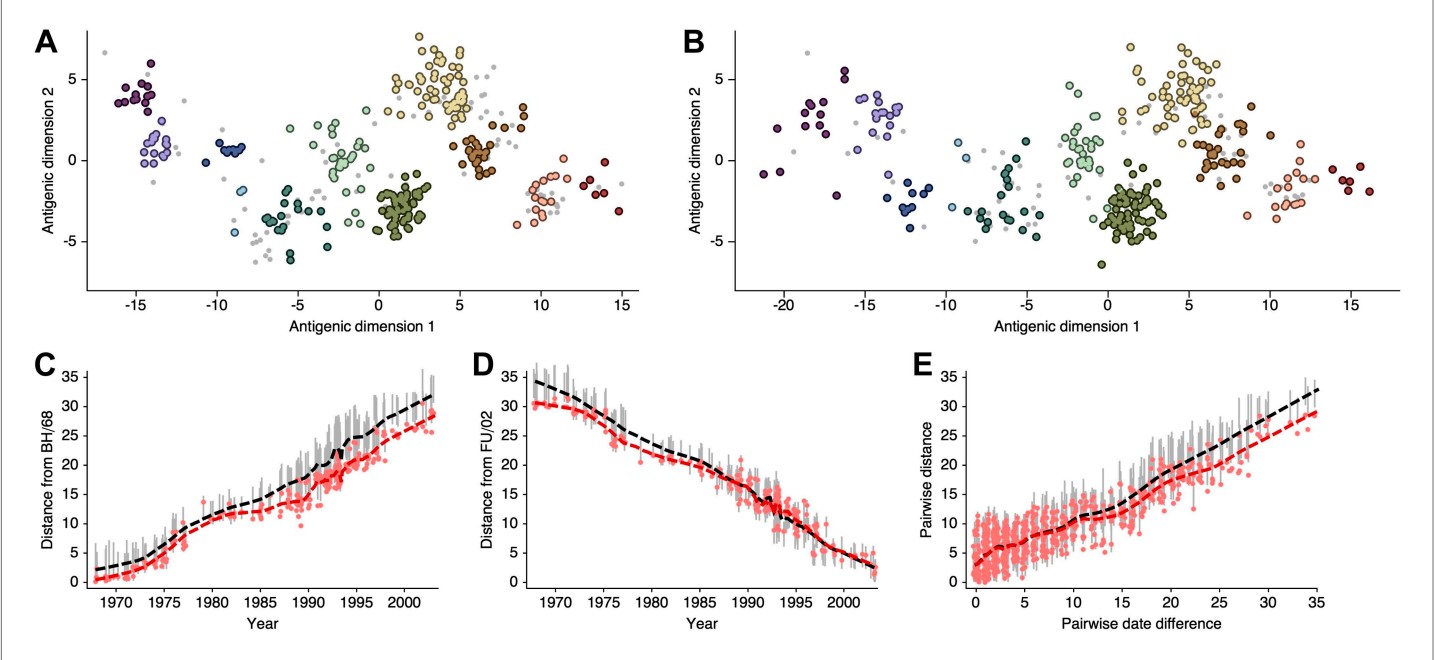

**Figure 13**. Comparison of A/H3N2 antigenic locations estimated by *Smith et al. (2004)* using MDS and an extended BMDS model that estimates serum and virus avidities. (**A**) MDS antigenic locations, reoriented so that the primary dimension lies on the *x*-axis rather than on the *y*-axis as in Figure 1 of *Smith et al. (2004)*. (**B**) A posterior sample of antigenic locations from a BMDS model that estimates virus avidity and serum potency. In (**A**) and (**B**), viruses are shown as colored circles, with color denoting antigenic cluster inferred by *Smith et al. (2004)*, and sera are shown as gray points. (**C**) Antigenic distances between A/Bilthoven/16,190/1968 and all other viruses determined for both methods. (**D**) Antigenic distances between A/Fujian/411/2002 and all other viruses determined for both methods. (**E**) Antigenic distances between 750 random pairs of viruses determined for both methods. In (**C**), (**D**), and (**E**) red points show distances for the MDS model and gray bars show the 95% credible interval of distances for the BMDS model, while the red dashed line shows a LOESS regression to MDS distances, and the black dashed line shows a LOESS regression to the BMDS distances. The BMDS model has a Uniform (−100, 100) prior on antigenic locations and virus avidities and serum potencies estimated in a hierarchical Bayesian fashion.

The following source data are available for figure 13:

**Source data 1**. This tab-delimited text file lists every virus and serum in *Figure 13*, including strain name, year of isolation, coordinates in antigenic dimensions 1 and 2, potency for sera and avidity for viruses.

## Genetic, antigenic and surveillance data

We compiled an antigenic dataset of hemagglutination inhibition (HI) measurements of virus isolates against post-infection ferret sera for influenza A/H3N2 by collecting data from previous publications (*Hay and Gregory, 2001*; *Smith et al., 2004*; *Russell et al., 2008*; *Barr et al., 2010*), NIMR vaccine strain selection reports for 2002 and 2008–2012 (*Hay et al., 2002*; *2008a*; *2009a*; *McCauley et al., 2010a*; *2010b*; *2011b*; *2012*) and the February 2011 VRBPAC report (*Cox, 2011*). We queried the Influenza Research Database (*Squires et al., 2012*) and the EpiFlu Database (*Bogner et al., 2006*) for HA nucleotide sequences by matching strain names, for example A/HongKong/1/1968, and only strains for which sequence was present were retained. If a strain had multiple sequences in the databases, we preferentially kept the IRD sequence and preferentially kept the longest sequence in IRD. Many strains had full length HA sequences, while other strains only possessed HA1 sequences. Sequences were aligned using MUSCLE v3.7 under default parameters (*Edgar, 2004*). This dataset had 2051 influenza isolates (present as either virus or serum in HI comparisons) dating from 1968 to 2011. However, the majority of isolates were present from 2002 to 2007. Because we are interested in longer-term antigenic evolution, we subsampled the data to have at most 20 virus isolates per year, preferentially keeping those isolates with more antigenic comparisons. We then kept only those serum isolates that are relatively informative to the antigenic placement of viruses, dropping serum isolates that are compared to four or fewer different virus isolates. This censoring left 402 virus isolates, 519 serum

isolates and 10,059 HI measurements. Each virus isolate was compared to an average of 21.9 serum isolates, and each serum isolate was compared to an average of 18.0 virus isolates.

Antigenic data for influenza A/H1N1 was collected from previous publications (*Kendal et al., 1978*; *Nakajima et al., 1979*; *Webster et al., 1979*; *Nakajima et al., 1981*; *Chakraverty et al., 1982*; *Pereira and Chakraverty, 1982*; *Cox et al., 1983*; *Daniels et al., 1985*; *Chakraverty et al., 1986*; *Raymond et al., 1986*; *Stevens et al., 1987*; *Donatelli et al., 1993*; *Hay and Gregory, 2001*; *Daum et al., 2002*; *McDonald et al., 2007*; *Barr et al., 2010*) and NIMR vaccine strain selection reports for 2002–2010 (*Hay et al., 2002, 2008a, 2009a*; *McCauley et al., 2010*; *Hay et al., 2003, 2004, 2005a, 2005b, 2006a, 2006b, 2007a, 2007b, 2008b*). The same procedure that was followed for A/H3N2 was repeated to match sequence data and to subsample antigenic comparisons. This procedure yielded 115 virus isolates, 77 serum isolates, and 1882 HI measurements over the course of 1977–2009. Each virus isolate was compared to an average of 10.0 serum isolates, and each serum isolate was compared to an average of 16.2 virus isolates.

Antigenic comparisons for influenza B/Victoria were collated from previous publications (*Rota et al., 1990*; *Hay and Gregory, 2001*; *Muyanga et al., 2001*; *Shaw et al., 2002*; *Ansaldi et al., 2004*; *Puzelli et al., 2004*; *Xu et al., 2004*; *Barr et al., 2006*; *Daum et al., 2006*; *Lin et al., 2007*) and vaccine strain selection reports for 2002–2012 (*Hay et al., 2002, 2008a, 2009a*; *McCauley et al., 2010a*; *2010b, 2011b, 2012*; *Hay et al., 2003, 2004, 2005a, 2005b, 2006a, 2006b, 2007a, 2007b*; *Gust et al., 2006*; *Hay et al., 2009a*; *McCauley et al., 2011b*). Here, the sequence matching and subsampling procedure yielded 179 virus isolates, 70 serum isolates and 2003 HI measurements over the course of 1986–2011. Each virus isolate was compared to an average of 6.5 serum isolates, and each serum isolate was compared to an average of 16.7 virus isolates.

Antigenic comparisons for influenza B/Yamagata were collected from previous publications (*Kanegae et al., 1990*; *Rota et al., 1990*; *Nakajima et al., 1992*; *Nerome et al., 1998*; *Hay and Gregory, 2001*; *Muyanga et al., 2001*; *Nakagawa et al., 2002*; *Shaw et al., 2002*; *Abed et al., 2003*; *Ansaldi et al., 2003, 2004*; *Matsuzaki et al., 2004*; *Puzelli et al., 2004*; *Xu et al., 2004*; *Barr et al., 2006*; *Daum et al., 2006*; *Lin et al., 2007*) and vaccine strain selection reports for 2002–2012 (*Hay et al., 2002, 2008a, 2009a*; *McCauley et al., 2010a*; *2010b, 2011b, 2012*; *Hay et al., 2003, 2004, 2005a, 2005b, 2006a, 2006b, 2007a, 2007b*; *Gust et al., 2006*; *Hay et al., 2009a*; *McCauley et al., 2011b*). For B/Yamagata, the matching and subsampling procedure resulted in 174 virus isolates, 69 serum isolates and 1962 HI measurements over the course of 1987–2011. Each virus isolate was compared to an average of 6.9 serum isolates, and each serum isolate was compared to an average of 17.3 virus isolates.

Surveillance data was obtained from the Centers of Disease Control and Prevention FluView Influenza Reports from the yearly summaries of influenza seasons 1997–1998 to 2010–2011 (*Centers for Disease Control and Prevention, 2012*). As an example, one report states 'collaborating laboratories in the United States tested 195,744 respiratory specimens for influenza viruses, 27,682 (14%) of which were positive. Of these, 18,175 (66%) were positive for influenza A viruses, and 9507 (34%) were positive for influenza B viruses. Of the 18,175 specimens positive for influenza A viruses, 7631 (42%) were subtyped; 6762 (87%) of these were seasonal influenza A (H1N1) viruses, and 869 (13%) were influenza A (H3N2) viruses'. In this case, we estimate the relative proportion of A/H3N2 of the four lineages as 0.66 × 0.13 = 0.09. Similar calculations were performed for A/H1N1, B/Vic and B/Yam.

## Implementation

Phylogenetic trees were estimated for A/H3N2, A/H1N1, B/Vic, and B/Yam using BEAST (*Drummond et al., 2012*) and incorporated the SRD06 nucleotide substitution model (*Shapiro et al., 2006*), a coalescent demographic model with constant effective population size and a strict molecular clock across branches. MCMC was run for 60 million steps and trees were sampled every 50,000 steps after allowing a burn-in of 10 million steps, yielding a total sample of 2000 trees. These trees were treated as a discrete set of possibilities when subsequently sampled in the BMDS analysis (*Pagel et al., 2004*). However, it would be possible to jointly sample from sequence data and serological data using these methods.

MCMC was used to sample virus locations $\mathbf{X}$, serum locations $\mathbf{Y}$, virus avidities $\mathbf{v}$, serum potencies $\mathbf{s}$, MDS precision $1/\varphi^2$, antigenic drift rate $\mu$, virus location precision $1/\sigma_x^2$, serum location precision $1/\sigma_y^2$, and phylogenetic tree $\tau$. MCMC chains were run for 500 million steps and parameter values sampled every 200,000 steps after a burn-in of 100 million steps, yielding a total of 2000 MCMC

samples. In all cases, when drift parameter $\mu$ was included the MCMC chain mixed well and arrived at the same estimated posterior distribution from different starting points. However, without drift parameter $\mu$, maps for A/H3N2 showed some degree of metastability, where some chains would converge on one solution and other chains would converge on a different solution. We favor models that include $\mu$, because its inclusion, in addition to correcting most identifiability issues, yields much improved mixing of antigenic locations.

There is some difficulty in summarizing posterior cartographic samples, as sampled virus and serum locations represent only relative quantities, and because of this, over the course of the MCMC, virus locations may shift. Our prior on virus and serum locations remove much of this issue, orienting the antigenic map along dimension 1 and fixing it to begin at the origin. However, local isometries are often still a problem. For example, in A/H3N2 the HK/68, EN/72 and VI/75 clusters may rotate in relation to other clusters. Consequently, it may be difficult to fully align MCMC samples using Procrustes analysis. For the present study, we take a simple approach and sample a single MCMC step and visualize the antigenic locations at this state (*Figure 2*, *Figure 3*). Then, for specific quantities of interest, like rate of antigenic drift and rate of diffusion at different points along the phylogeny, we calculate the quantity across MCMC samples to yield an expectation and a credible interval. This approach accurately characterizes uncertainty that may be hidden in an analysis of a single antigenic map.

We summarize diffusion paths of viral lineages (*Figure 2*, *Figure 3*) by taking each virus and reconstructing x and y locations along antigenic dimensions 1 and 2 backward through time. We use MCMC to sample tip locations, but when outputting trees sample internal node locations using a peeling algorithm as described in *Pybus et al. (2012)*. Thus, after the MCMC is finished, we have a posterior sample of 2000 trees each tagged with estimated tip locations and internal node locations. We postprocessed each posterior tree by conducting a linear interpolation between parent–child node locations to arrive at x and y values at intervals of 0.05 years for each virus. Then, for each interval, x and y values are averaged across the sample of posterior trees. We draw lines between these locations to approximate mean posterior diffusion paths. As virus lineages coalesce backwards through time down the phylogeny these diffusion paths will also coalesce.

## Comparison with previous results

Here, we attempt to compare antigenic locations inferred by our BMDS model to antigenic locations previously inferred by the error minimization methods of *Smith et al. (2004)*, referred to here as antigenic cartography by MDS. For this comparison, we use exactly the same HI data used to produce the results in *Smith et al. (2004)*, consisting of 273 virus isolates, 79 serum isolates and a total of 4252 HI measurements taken between 1968 and 2003. We begin with a BMDS analog of the antigenic model used in *Smith et al. (2004)*, where serum potencies are taken as the maximum titer of a particular ferret serum and the expected $\log_2$ drop in HI titer is proportional to Euclidean distance between virus and serum locations. To bring models into further alignment, we use a Uniform ($-100$, $100$) distribution over virus locations and serum locations. Unsurprisingly, we find that this BMDS model produces results that are strongly congruent with MDS results (*Figure 10*, *Figure 10—source data 1*). Antigenic cluster locations are consistent between methods (*Figure 10A–B*) and antigenic distances between pairs of viruses are consistent between temporally similar and temporally divergent viruses (*Figure 10C–E*), suggesting that the resulting maps are consistent at both local and global scales. Credible intervals of antigenic distances for the BMDS model remain narrow across the temporal spectrum (*Figure 10C–E*), implying a fair degree of rigidity to the map.

*Smith et al. 2004* show that there exist at least two solutions in their assignment of antigenic locations, involving the rotation of clusters HK/68, EN/72 and VI/75 (shown in Figure S2 of [*Smith et al., 2004*]). We observe the same metastable behavior in our analysis; some MCMC chains converge on the solution shown in *Figure 10B*, while other MCMC chains converge on the alternative solution shown in *Figure 11B*. The distribution of likelihood values appears highly similar between these two solutions, suggesting that they represent global optima. The rotation of the HK/68, EN/72, and VI/75 clusters creates a map that bends slightly, so that temporally distant viruses appear closer in the rotated solution than in the original solution (*Figure 11C–E*, *Figure 11—source data 1*). In this case, it is clear that the solutions are locally consistent between viruses up to ~15 years divergent, even if there is some degree of global flexibility.

As discussed in the main text, the presence of multiple optima with different degrees of 2D curvature implies an issue of identifiability; the HI likelihood model alone cannot distinguish between these

possibilities. Because of this issue, and to more easily estimate rates of antigenic drift, we include a model of systematic drift in antigenic location that favors linear movement in the antigenic map. We find that including this drift prior on antigenic locations removes the problem of identifiability. Antigenic locations produced by this model remain locally consistent with MDS results between viruses ~15 years divergent, but global comparisons show that this BMDS model has partitioned more variance to the first antigenic dimension (*Figure 12*, *Figure 12—source data 1*). We additionally find that including the drift prior on antigenic locations often results in greater predictive power, with a slight improvement of test error for the A/H1N1, B/Vic, and B/Yam datasets (*Table 1*).

Our final BMDS model (model 9, *Table 1*) differs from antigenic model used by *Smith et al. (2004)* in including temporally- and phylogenetically-informed priors on antigenic locations and also in estimating serum and virus avidities. Here, we investigate the impact on antigenic locations of estimating virus avidity and serum potency in the BMDS model. To isolate this difference, we use a Uniform (−100, 100) prior on antigenic locations. Surprisingly, estimating virus avidity and serum potency results in a more linear antigenic map (*Figure 13*, *Figure 13—source data 1*), resembling the appearance of the map incorporating the antigenic drift prior, while preserving local consistency. We generally observe congruence between MDS and BMDS antigenic locations for viruses less than ~10 years divergent (*Figure 13E*). However, specific viruses may be affected, for instance A/Bilthoven/16,190/1968 (*Figure 13C*), which appears more distant from all other viruses when serum and virus avidities are included.

In this dataset, viruses 15 or more years divergent always yield threshold titers, and hence, their relative locations must be indirectly inferred rather than through direct comparison. This may explain why we observe local consistency between models at scales less than ~15 years, but some degree of global inconsistency. Still, these results suggest that, when making local comparisons, such as those used to calculate year-to-year antigenic drift (*Figure 3*), outcomes are expected to be robust to many model particulars.

## Availability

Source code implementing the cartographic models has been made fully available as part of the software package BEAST (*Drummond et al., 2012*), and can be downloaded from its Google code repository (http://code.google.com/p/beast-mcmc/). More details on implementing these models can be found at https://github.com/trvrb/flux/tree/master/example-xmls. Incidence data and HI data used in this analysis is archived with Dryad (doi: 10.5061/dryad.rc515).

## Acknowledgements

We thank Richard Reeve, Dan Haydon, and Simon Frost for insights on antigenic modeling and MDS. We acknowledge the laboratories that provided sequences to EpiFlu database: Centers for Disease Control and Prevention (USA), Chinese Center of Disease Prevention and Control, Hospital Clinic of Barcelona, National Institute of Hygiene of Morocco, National Institute of Infectious Diseases (Japan), National Institute for Medical Research (UK), Norwegian Institute of Public Health, Swedish Institute for Infectious Disease Control, Victorian Infectious Diseases Reference Laboratory (Australia).

## Additional information

### Funding

| Funder | Grant reference number | Author |
| --- | --- | --- |
| Royal Society | Newton International Fellowship | Trevor Bedford |
| National Institutes of Health | R01 HG006139 | Marc A Suchard |
| National Institutes of Health | R01 AI107034 | Marc A Suchard |
| National Science Foundation | DMS126153 | Marc A Suchard |
| National Science Foundation | IIS1251151 | Marc A Suchard |
| European Commission | 278433-PREDEMICS | Marc A Suchard, Philippe Lemey, Andrew Rambaut |

| Funder | Grant reference number | Author |
|---|---|---|
| European Commission | 260864 | Marc A Suchard, Philippe Lemey, Andrew Rambaut |
| National Science Foundation | EF-0423641 | Marc A Suchard, Philippe Lemey, Andrew Rambaut |
| Medical Research Council | U117512723 | Victoria Gregory, Alan J Hay, John W McCauley |
| Wellcome Trust | 092807 | Andrew Rambaut |
| European Commission | 223498-EMPERIE | Colin A Russell, Derek J Smith |
| European Commission | 278976-ANTIGONE | Colin A Russell, Derek J Smith |
| Human Frontier Science Program | P0050/2008 | Colin A Russell, Derek J Smith |
| Wellcome Trust | 087982AIA | Colin A Russell, Derek J Smith |
| National Institutes of Health | DP1-OD000490-01 | Colin A Russell, Derek J Smith |
| National Institutes of Health | HHSN266200700010C | Colin A Russell, Derek J Smith |
| Royal Society | University Research Fellowship | Colin A Russell |

The funders had no role in study design, data collection and interpretation, or the decision to submit the work for publication.

## Author contributions

TB, DJS, AR, Conception and design, Acquisition of data, Analysis and interpretation of data, Drafting or revising the article, Contributed unpublished essential data or reagents; MAS, PL, Conception and design, Analysis and interpretation of data, Drafting or revising the article, Contributed unpublished essential data or reagents; GD, Acquisition of data, Analysis and interpretation of data, Drafting or revising the article; VG, Acquisition of data, Contributed unpublished essential data or reagents; AJH, JWM, CAR, Acquisition of data, Analysis and interpretation of data, Drafting or revising the article, Contributed unpublished essential data or reagents

# Additional files

### Major dataset

The following dataset was generated:

| Author(s) | Year | Dataset title | Dataset ID and/or URL | Database, license, and accessibility information |
|---|---|---|---|---|
| Bedford T, Suchard MA, Lemey P, Dudas G, Gregory V, Hay AJ, McCauley JW, Russell CA, Smith DJ, Rambaut A | 2013 | Data from: Integrating influenza antigenic dynamics with molecular evolution | http://dx.doi.org/10.5061/dryad.rc515 | Available at Dryad Digital Repository under a CC0 Public Domain Dedication. |

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
