## [Decision Letter]

[Editors’ note: a previous version of this study was rejected after peer review, but the authors submitted for reconsideration. The two decision letters after peer review are shown below.]

Thank you for choosing to send your work entitled “Integrating influenza antigenic dynamics with molecular evolution” for consideration at *eLife*. Your article has now been peer reviewed and we regret to inform you that your work will not be considered further for publication at this stage. Your submission has been evaluated by a Senior editor and 2 reviewers, and the decision was reached after discussions between the reviewers.

The reviewers and the Senior editor found this extension of antigenic cartography most interesting, both methodologically with the introduction of a Bayesian framework, and from the inclusion of more clades and H1 and B data. The central issue, however, is that we are not convinced of a causal link between drift and incidence. The correlation between clades could simply be driven by the fact that H3N2 has more drift and higher incidence. Secondly, the correlation across years within a given clade could be driven by sampling biases. For example, if H1N1 is reported more than usual in a given year, it will typically have more isolates subjected to HI assays, which by itself might make it appear to have drifted more. (We note, however, that this issue might be addressable if the authors were to draw a random sample of the isolates such that the number per year were equal.)

In sum, we are concerned that the analysis suggests a bigger connection between drift and incidence than actually exists. It is not clear if there is correlation within clades (these plots were not shown), and the across-clade correlation could be at least partly an artifact of the pooling of clades. In addition, the across-clade and within-clade effects could be amplified by sampling bias. Given, however, the extreme importance of this work to understanding influenza evolution and vaccine selection and the potential to provide a quantitative assessment of the association between drift (as measured by cartography) and incidence, the reviewers and editor would be willing to consider a suitably revised submission with major revisions that convinced us that the correlation is true or that there is perhaps no correlation between drift and incidence (or conceivably that antigenic cartography is not a good way to measure drift).

*Reviewer #1 comments*:

This paper is exceptional in many ways, but also contains some flaws that need to be addressed by revisions. If these flaws are addressed, I would be highly supportive of publication.

This paper deals with the antigenic evolution of influenza. A foundational paper in Science in 2004 by Smith et al introduced “antigenic cartography,” the technique used to monitor antigenic drift and select the influenza vaccine. This paper makes a clear methodological improvement on the original antigenic cartography technique. The new approach integrates phylogenetics, allowing inclusion of viral sequences and isolation dates. This paper also shows that accounting for virus and serum effects improves cartography. As the authors briefly note, the new method is also extensible to evolutionary questions. All of these improvements are made using a freely available phylogenetic software program that is the standard in the field. These methodological improvements would immediately merit publication in a technique-specific journal.

In addition, this paper presents biological results that should interest virologists, public health officials, and evolutionary biologists. While these results do not really alter the current understanding of influenza antigenic evolution, they extend this understanding from H3N2 to other clades, provide a more detailed analysis than has previously been performed for any clade, and quantify factors that have previously been characterized only qualitatively.

I do, however, have some major critiques:

• It is difficult to understand the biological results without grasping technical aspects of the computational approach. In my specific comments, I suggest some changes.

• I do not concur that Figure 5 indicates “antigenic drift drives incidence rates.” There are problems with making the correlations in Figure 5 across all four clades, since the clades have different average rates of drift and incidence. The conclusion would be supported if there was correlation between incidence and rate within clades, or if the correlation across clades exceeded the correlation of the clade averages. But as it stands, Figure 5's correlations are just due to the fact that the four clades differ in the average rates, which does not imply that drift drives incidence. This paper is worth publishing even if a proper analysis fails to find that drift drives incidence, because currently the relationship is widely assumed without quantitative analysis. However, it is important to get this right as it forms the basis for influenza vaccine selection. I describe my concerns in more detail in the specific comments.

• The authors need to explain where they got the incidence data, how they chose the date ranges, specify when they are using relative versus absolute incidence, and provide raw data as supporting material. I describe my concerns in more detail in the specific comments.

• The improvement from inclusion of serum effects deserves comment and explanation if possible (are the effects due to the serum source)? Table 1 should include a model with virus effects but no serum effects. I describe my concerns in more detail in the specific comments.

Major points in more detail:

1) I have concerns about Figure 5, which the authors use to support their claim that “antigenic drift drives incidence rates.” The authors first establish the well-known fact that H3N2 incidence is highest, followed by H1N1 and then B. They also find that relative incidence is correlated with the rate of antigenic drift. However, this does not imply causality in terms of the higher average drift driving the higher average incidence. The authors do not claim any causality in this correlation between average drift and incidence in the four clades, and I am therefore fine with this part of the analysis. The problems arise in Figure 5. The authors find a significant correlation between drift and incidence taken over all years and clades, and then use this to argue that drift drives incidence. I would agree with this conclusion if higher drift was associated with higher incidence with a correlation that is better than that obtained simply by comparing the averages of the four clades. But I don't think this is true. In Figure 5, the authors are averaging over all clades, making the correlation in 5A a trivial consequence of the fact that H3N2 has higher average incidence and drift. Even if the rate of drift and incidence were constant year-over-year for each clade (and simply differed among clades), the authors would observe the correlation that they report. Put another way; imagine that every year the drift and incidence in each clade was equal to its average. Then Figure 5 would in effect correspond to simply recalculating the correlation between average drift and incidence in a scenario where each point is included as many times as there are years being examined. The more years that were included the higher the P-value would get because the number of data points would increase, but really the single actual trend (that H3N2 is higher in average drift and incidence) is just being counted multiple times. In order to draw a conclusion stronger that drift drives incidence, the authors need to show that incidence and drift are correlated within each clade individually, or else standardize the incidence so that the mean and variance for each individual clade is zero and one. It is not clear to me that there is any correlation between the drift and incidence within any of the individual clades. A similar problem affects Figure 5 - for example, since H3N2 has the highest drift, it will always tend to be associated with lower drift in other clades since the other clades inherently have lower average drift. But this does not imply direct interference between clades. I am open to further discussion about this point, but I do not see how Figure 5 implies that antigenic drift drives incidence, or that there is dynamical interference across clades. By the way, I do not think that a lack of correlation would be a disqualifying concern in terms of publication. Finding no evidence that antigenic drift drives incidence is just as important as finding that it does. But the claim needs to be correct.

2) The incidence data are poorly explained. In the first paragraph of the Epidemiological consequences of antigenic drift section, the authors use relative incidence. In the next paragraph, are they switching to absolute incidence? This is not clear. If they are switching to absolute incidence, I don't think they ever explain in the Methods where that data comes from. In either case, what are the justifications for using absolute versus relative incidence? Also, why do they not analyze incidence prior to 1998/1999? Finally, because each influenza season spans two calendar years, why do they not break up the data by season rather than year?

3) An interesting result is that estimating the serum effects improves the performance a great deal (more than including the phylogenetic data, in fact!). It suggests that there may be systematic differences between the sera. Can this be attributed to the serum itself (such as whether it is goat versus ferret), or to the particular lab or study? Could the authors correlate the estimated serum effects back to the originating source for that serum to check? As a related point, I was surprised that inclusion of the virus effects led to only minor improvement, since prior work (http://www.ncbi.nlm.nih.gov/pubmed/19900932) suggests that some viruses are generally more resistant to HI than others. However, the virus effects are only included in a model (#9 in Table 1) where the serum effects are already included, and it is possible to imagine that including just one of these terms somehow helps correct for both effects. Can the authors include a tenth model in Table 1 that includes the virus effects but not the serum effects? This would clarify if serum effects are actually more important than virus effects, or simply whether inclusion of either one individually is more helpful than both together.

*Reviewer #2 comments*:

The authors extend earlier work on “antigenic cartography” — that is, low-dimensional embeddings of viral strains using dimensionality reduction techniques, in order to quantify antigenic groupings of viral isolates. What is novel here compared to prior work is the introduciton of a Bayesian framework for this mapping process, the observation that antigenic drift within a clade correlates with clade-specific disease incidence each season, and anti-correlates with drift in other clades. The authors also introduce a phenomonological model of antigenic variation based on diffusion, and interpret antigenic drift in terms of fitted diffusion coefficiants. Finally, the authors provide antigenic maps for more clades of influenza (H3N2, H1N1, B/Vic, and B/Yam) than in previous publications.

Understanding antigenic drift of influenza in quantitative terms is clearly of great practical importance, and also an interesting intellectual problem in its own right. And it is nice to see antigenic maps for so many viral clades. However, I do not feel that the technical advances in “mapping” strains introduced here provide us much more confidence in the antigenic locations of strains than prior techniques do. Indeed, the authors themselves note that the average predictive error of this new methods is comparable to that of Smith et al. (2004). What might be more valuable, perhaps, are the results relating inferred antigenic drift and influenza incidence each season, by clade. But these results are quite possibly driven by sampling biases (see below), and I am not really convinced of their validity — at least not without serious efforts to remove such sampling biases. Finally, the discussion of a “diffusion” model of drift was extremely confusing to me, and the fitted diffusion coefficients do not have a clear meaning or interpretation.

Serious concerns follow:

1) The authors do not seem to discuss a problematic aspect of their results relating inferred drift to influenza disease incidence — namely that biased sampling of clades, in each season, could cause the observed correlations. If clades were sampled for HI assays unequally in each season, as I suspect, then this fact alone might cause the trend towards apparently more drift in the clade that was sampled most intensively in each season (and to which most ILI was attributed). This is an obvious possible confound for the analysis presented in Figure 4, and it should be ruled out by simulation of constant true rates of antigenic drift, variable sampling proportions in each season, and reconstructed drift using BMDS.

2) Likewise, in Figure 5, the negative correlation between drift in different clades in each season could, again, be driven by variation in the number of strains per clade that were subjected to HI each season. The authors could rule this out either by simulation or, more simply, be repeating their analysis after subsampling an equal number of isolates per clade each season for inclusion into the BMDS.

3) I found the section fitting diffusion coefficients (in different branches) to the antigenic maps confusing — if only because strict diffusion (in one or two dimensions) has no advection term and therefore no tendency to move in one direction or another; whereas the data clearly show an advective tendency (moving to the “right” over time in antigenic dimension 1). I presume the authors attempt to deal with this by imposing a prior (eq. 12) for the location of a virus based on its location in a molecular phylogeny. But this would seem to conflate antigenic drift, which the authors wish to quantify using a diffusion coefficient, with genetic mutations (many of which are driven by selection for antigenic escape). In other words, this hybrid approach of assuming a prior based on a strain's genotype, and then fitting a diffusion coefficient to summarize strength of drift, does not seem to be a pure measure of drift. What exactly the fitted diffusion coefficient means is extremely unclear to me, in any formal sense, as the procedure is so ad-hoc.

4) How do the antigenic maps shown compare to BMDS applied to pairwise AA hamming distances alone? How many serious discrepancies that are outside the confidence intervals on map reconstruction? In other words, the authors should demonstrate that the HI data actually provide substantively different results for antigenic grouping than genetic data do alone. This was an important contention of Smith et al. (2004), but it was based on the position of only a handful of viruses in that paper. Do the authors have more convincing evidence in this study that the antigenic data are providing substantial extra information beyond the genetic data?

[Editors’ note: what now follows is the decision letter after the authors submitted for further consideration.]

Thank you for resubmitting your work entitled “Integrating influenza antigenic dynamics with molecular evolution” for consideration at *eLife*. Your submission has been evaluated by a Senior editor and the original two reviewers.

Both reviewers were pleased with the revised manuscript and only a small number of issues remain to be addressed before acceptance. These are: whether virus effects could be due to avidity, clarifying the meaning of tau in Table 2, expanding on the rationale for subsampling in the Punctuated evolution section, the apparent discrepancy between Figure 4 and its legend, eliminating the orphan paragraph described in point #3 below, and a paragraph in the Discussion summarizing the main biological results. These points are explained in the reviewers’ comments below.

1) I like how the authors have included “virus effects” alone in the new Table 1. At least one “virus effect” could be overall receptor avidity, and Plotkin and Hensley have a recent paper (Journal of Virology, 87:9904) showing that avidity can influence antigenic clustering. It might be worthwhile to include a sentence on the possibility that “virus effects” could be a manifestation of avidity?

2) In Table 2 and the related discussion, the authors discuss the scaled effective population size N_e_ * τ. They never define what tau represents, and I don't think it is safe to assume that the reader will know this — I certainly don't. More interpretation here would be helpful.

3) In the “Punctuated evolution and its epidemiological consequences” section, the part about subsampling to look at whether drift associates with number of isolates should be expanded. I would suggest starting a new paragraph at the current “We test to see...” sentence that briefly explains the rationale for why this test is being done (rationale articulated by other reviewer in original critiques). I also think that it would be nice to have a table or figure somehow representing the actual results of this analysis.

4) Figure 4 legend refers to “The mean posterior scaled effective populations... is shown for each virus.” These are not actually shown in Figure 4, at least not in a way that is obvious to me.

5) My comments where the original reason that the authors removed the argument for the across lineage correlation in incidence. However, now ‘Punctuated evolution and its epidemiological consequences’ section has this orphan paragraph beginning “Although the general correlation between rate of antigenic drift...” This paragraph doesn't seem to make sense in the context of the presented data any more, as the paper no longer has any information about across lineage correlations, so it can't even be seen what they are saying may not be causal. I think this paragraph either needs to be dramatically expended or probably better eliminated. Maybe the previous paragraph could then just get a wrap-up sentence interpreting the results to suggest a strong relationship between drift and incidence within each lineage.

6) Although the Conclusion is fine, I feel that it might benefit from a paragraph summarizing the main biological results as regards different rates of drift in lineages and the correlation between drift and incidence within lineages. These biological results are not really mentioned in the current Conclusion.

---

## [Author Response]

[Editors’ note: the author responses to the first round of peer review follow.]

Reviewer #1 comments:

*1) I have concerns about*
Figure 5*, which the authors use to support their claim that “antigenic drift drives incidence rates.” The authors first establish the well-known fact that H3N2 incidence is highest, followed by H1N1 and then B. They also find that relative incidence is correlated with the rate of antigenic drift. However, this does not imply causality in terms of the higher average drift driving the higher average incidence. The authors do not claim any causality in this correlation between average drift and incidence in the four clades, and I am therefore fine with this part of the analysis. The problems arise in*
Figure 5*. The authors find a significant correlation between drift and incidence taken over all years and clades, and then use this to argue that drift drives incidence. I would agree with this conclusion if higher drift was associated with higher incidence with a correlation that is better than that obtained simply by comparing the averages of the four clades. But I don't think this is true. In*
Figure 5*, the authors are averaging over all clades, making the correlation in 5A a trivial consequence of the fact that H3N2 has higher average incidence and drift. Even if the rate of drift and incidence were constant year-over-year for each clade (and simply differed among clades), the authors would observe the correlation that they report. Put another way; imagine that every year the drift and incidence in each clade was equal to its average. Then*
Figure 5
*would in effect correspond to simply recalculating the correlation between average drift and incidence in a scenario where each point is included as many times as there are years being examined. The more years that were included the higher the P-value would get because the number of data points would increase, but really the single actual trend (that H3N2 is higher in average drift and incidence) is just being counted multiple times. In order to draw a conclusion stronger that drift drives incidence, the authors need to show that incidence and drift are correlated within each clade individually, or else standardize the incidence so that the mean and variance for each individual clade is zero and one. It is not clear to me that there is any correlation between the drift and incidence within any of the individual clades. A similar problem affects*
Figure 5
*- for example, since H3N2 has the highest drift, it will always tend to be associated with lower drift in other clades since the other clades inherently have lower average drift. But this does not imply direct interference between clades. I am open to further discussion about this point, but I do not see how*
Figure 5
*implies that antigenic drift drives incidence, or that there is dynamical interference across clades. By the way, I do not think that a lack of correlation would be a disqualifying concern in terms of publication. Finding no evidence that antigenic drift drives incidence is just as important as finding that it does. But the claim needs to be correct*.

This is a very astute criticism. We see how differing overall incidence between clades could have given an artifactual signal in our previous year-to-year drift vs incidence comparisons. In this revision, we have reanalyzed the data in the way suggested; we show that year-to-year antigenic drift and incidence are correlated within each clade individually, arriving at correlation coefficients of 0.51, 0.29, 0.44 and 0.14 for A/H3N2, A/H1N1, B/Vic and B/Yam respectively. None of these correlations are significant on their own, however observing four correlation coefficients of this magnitude is highly unlikely under a null model derived from bootstrap permutations (p = 0.018). Because the increase in incidence tends to follow periods of pronounced antigenic drift, we conclude that there appears to be a causal relationship between antigenic change and increased incidence.

However, in redoing this analysis on a lineage-by-lineage basis, we lost much of the signal for interference between lineages. We think there may still be something there, but the nuanced analysis that this issue deserves seems beyond the scope of the paper. We have decided to instead drop the discussion of interference between lineages.

*2) The incidence data are poorly explained. In the first paragraph of the Epidemiological consequences of antigenic drift section, the authors use relative incidence. In the next paragraph, are they switching to absolute incidence? This is not clear. If they are switching to absolute incidence, I don't think they ever explain in the Methods where that data comes from. In either case, what are the justifications for using absolute versus relative incidence*?

In revising this section in the manuscript, we have made the source of the incidence data and its construction much more clear. In addition, we have removed the relative vs absolute incidence confusion, sticking instead with one measure: ILI × proportion of viral isolates attributable to a lineage.

*Also, why do they not analyze incidence prior to 1998/1999*?

The 1998/1999 season is the earliest we have that properly distinguishes B/Vic and B/Yam in the CDC data.

*Finally, because each influenza season spans two calendar years, why do they not break up the data by season rather than year*?

This would be much preferable. However, many virus isolates used in this analysis lack temporal resolution beyond the year of sampling. Because of this, we decided to stick with just year of isolation for the present analysis.

*3) An interesting result is that estimating the serum effects improves the performance a great deal (more than including the phylogenetic data, in fact!). It suggests that there may be systematic differences between the sera. Can this be attributed to the serum itself (such as whether it is goat versus ferret), or to the particular lab or study? Could the authors correlate the estimated serum effects back to the originating source for that serum to check*?

We would suggest that a serum with a larger effect contains a more concentrated and active set of antibodies than a serum with a smaller effect. The increased number of antibodies could be due to experimental variation in serum extraction and processing or due to variation in immune response and timing between ferrets, i.e., some ferrets may mount an immediate and strong immune response, while others may mount a weaker response. This causes variation in overall strength of the serum. We try to control for the overall strength when looking at patterns of cross-reactivity through the use of ‘serum effects’. Some differences in serum effects could be tracked down to particular studies, but this work is beyond the scope of the current study. Here, we’ve essentially treated serum effect as a nuisance parameter. We’ve revised the text to discuss this as well as describe why estimating serum effects improves model performance over fixing serum effects.

*As a related point, I was surprised that inclusion of the virus effects led to only minor improvement, since prior work (**http://www.ncbi.nlm.nih.gov/pubmed/19900932**) suggests that some viruses are generally more resistant to HI than others. However, the virus effects are only included in a model (#9 in Table 1) where the serum effects are already included, and it is possible to imagine that including just one of these terms somehow helps correct for both effects. Can the authors include a tenth model in Table 1 that includes the virus effects but not the serum effects? This would clarify if serum effects are actually more important than virus effects, or simply whether inclusion of either one individually is more helpful than both together*.

The inclusion of ‘virus effects’ was meant to capture exactly this observation, that some viruses may be generally more resistant to HI. To further assess their importance, we followed this suggestion and included test error for a model that estimates virus effects, but leaves serum effects fixed at maximum titers. In this case, we observe similar, but not quite as substantial, improvements to test error as the model that estimates serum effects, but does not include virus effects. From this observation, we conclude that estimating either virus or serum effects both improve the model.

Reviewer #2 comments:

*1) The authors do not seem to discuss a problematic aspect of their results relating inferred drift to influenza disease incidence—namely that biased sampling of clades, in each season, could cause the observed correlations. If clades were sampled for HI assays unequally in each season, as I suspect, then this fact alone might cause the trend towards apparently more drift in the clade that was sampled most intensively in each season (and to which most ILI was attributed). This is an obvious possible confound for the analysis presented in*
Figure 4*, and it should be ruled out by simulation of constant true rates of antigenic drift, variable sampling proportions in each season, and reconstructed drift using BMDS*.

We tested to see if there might be an bias in the estimates of year-to-year anti- genic drift based on number of virus isolates collected by correlating correlating drift between years i and i − 1 against the ratio of the number of isolates from year i to the number of isolates from year i−1. We found no significant correlation (Pearson’s r = −0.01, p = 0.411), suggesting our results to be unbiased with regard to sample count. We have included this analysis in the manuscript.

*2) Likewise, in*
Figure 5*, the negative correlation between drift in different clades in each season could, again, be driven by variation in the number of strains per clade that were subjected to HI each season. The authors could rule this out either by simulation or, more simply, be repeating their analysis after subsampling an equal number of isolates per clade each season for inclusion into the BMDS*.

The above analysis applies to this correlation as well.

*3) I found the section fitting diffusion coefficients (in different branches) to the antigenic maps confusing — if only because strict diffusion (in one or two dimensions) has no advection term and therefore no tendency to move in one direction or another; whereas the data clearly show an advective tendency (moving to the “right” over time in antigenic dimension 1). I presume the authors attempt to deal with this by imposing a prior (eq. 12) for the location of a virus based on its location in a molecular phylogeny. But this would seem to conflate antigenic drift, which the authors wish to quantify using a diffusion coefficient, with genetic mutations (many of which are driven by selection for antigenic escape). In other words, this hybrid approach of assuming a prior based on a strain's genotype, and then fitting a diffusion coefficient to summarize strength of drift, does not seem to be a pure measure of drift. What exactly the fitted diffusion coefficient means is extremely unclear to me, in any formal sense, as the procedure is so ad-hoc*.

We agree that applying the standard measure of diffusion coefficient D based on displacement vs time, without accounting for advection was problematic. We have substantial revised the text to focus on direct parameter inference of drift parameter μ and diffusion volatility parameter σ_x_, properly separating advection and volatility in the diffusion process. The section “Antigenic evolution across influenza lineages” most reflects these changes, including the addition of a new Table 2.

*4) How do the antigenic maps shown compare to BMDS applied to pairwise AA hamming distances alone? How many serious discrepancies that are outside the confidence intervals on map reconstruction? In other words, the authors should demonstrate that the HI data actually provide substantively different results for antigenic grouping than genetic data do alone. This was an important contention of Smith et al. (2004), but it was based on the position of only a handful of viruses in that paper. Do the authors have more convincing evidence in this study that the antigenic data are providing substantial extra information beyond the genetic data*?

We have revised the manuscript to include a thorough analysis of the relation- ship between genetic distance and antigenic distance. This is new Figure 1. Rather than constructing BMDS maps with pairwise AA distances, we take a more direct approach and compare pairwise antigenic distance from a BMDS model using only HI data (model 2 in Table 1) and pairwise genetic distances. We examine both pairwise AA distance and pairwise phylogenetic distance, as phylogenetic distance was the basis for our evolutionary diffusion model. We find fairly modest correlations in most cases case (between 0.10 for B/Yam and 0.68 for A/H3N2). Genetic data provides some predictive power for antigenic distance, but is a rather weak predictor alone. This was a big part of our rationale for attempting a joint genetic/antigenic model.

[Editors’ note: the author responses to the re-review follow.]

*1) I like how the authors have included “virus effects” alone in the new Table 1. At least one “virus effect” could be overall receptor avidity, and Plotkin and Hensley have a recent paper (Journal of Virology, 87:9904) showing that avidity can influence antigenic clustering. It might be worthwhile to include a sentence on the possibility that “virus effects” could be a manifestation of avidity*?

We appreciate this suggestion. Including the biological explanation for why we observe virus effects in the form of decreased or increased overall HI reactivity is very important. We’ve revised this section to include references to virus avidity and relabeled ‘virus effects’ to ‘virus avidities’. This has the additional benefit of being more transparent of a term than the opaque ‘effect.’

With the change from ‘virus effect’ to ‘virus avidity’ made, we chose to make a similar biological realignment of ‘serum effect’ to ‘serum potency’, i.e. some sera have higher potency than other sera, allowing them to inhibit hemagglutination at lower concentrations than other sera. The use of potency here is meant to align with the neutralizing antibody literature that distinguishes neutralization potency from neutralization breadth. Antigenic cartography has not traditionally measured breadth of hemagglutination inhibition.

*2) In Table 2 and the related discussion, the authors discuss the scaled effective population size N*_*e*_
** τ. They never define what tau represents, and I don't think it is safe to assume that the reader will know this — I certainly don't. More interpretation here would be helpful*.

We’ve revised the manuscript to clarify the definition and interpretation of both Ne and τ.

*3) In the “Punctuated evolution and its epidemiological consequences” section, the part about subsampling to look at whether drift associates with number of isolates should be expanded. I would suggest starting a new paragraph at the current “We test to see...” sentence that briefly explains the rationale for why this test is being done (rationale articulated by other reviewer in original critiques). I also think that it would be nice to have a table or figure somehow representing the actual results of this analysis*.

We’ve expanded out this sentence to a paragraph following the paragraph on the year-to-year drift vs incidence correlation. We give rationale for the test, explain the bootstrap procedure, give p-values for both combined and separate analyses across lineages and provide a figure showing a scatterplot of the data.

*4)*
Figure 4
*legend refers to “The mean posterior scaled effective populations... is shown for each virus.” These are not actually shown in*
Figure 4*, at least not in a way that is obvious to me*.

Thank you for catching this. Estimates of N_e_τ had been included in Figure 4, but were moved to Table 2 and the Figure 4 legend was not updated accordingly. This has been fixed to leave just the estimates in Table 2.

*5) My comments where the original reason that the authors removed the argument for the across lineage correlation in incidence. However, now ‘Punctuated evolution and its epidemiological consequences’ section has this orphan paragraph beginning “Although the general correlation between rate of antigenic drift...” This paragraph doesn't seem to make sense in the context of the presented data any more, as the paper no longer has any information about across lineage correlations, so it can't even be seen what they are saying may not be causal. I think this paragraph either needs to be dramatically expended or probably better eliminated. Maybe the previous paragraph could then just get a wrap-up sentence interpreting the results to suggest a strong relationship between drift and incidence within each lineage*.

We agree with this advice. We have replaced this paragraph with a wrap-up sentence as suggested.

*6) Although the Conclusion is fine, I feel that it might benefit from a paragraph summarizing the main biological results as regards different rates of drift in lineages and the correlation between drift and incidence within lineages. These biological results are not really mentioned in the current Conclusion*.

We have revised the Conclusion to include more discussion of the biological results.